# ANKEF1 is a key axonemal component essential for murine sperm motility and male fertility

Shuntai Yu[1,2†], Guoliang Yin[3†], Peng Jin[2], Weilin Zhang[2], Yingchao Tian[2], Xiaotong Xu[2], Tianyu Shao[1,2], Yushan Li[4], Fei Sun[3], Yun Zhu[3*], Fengchao Wang[2,5*]

[1]Academy for Advanced Interdisciplinary Studies, Peking University, Beijing, China; [2]National Institute of Biological Sciences (NIBS), Beijing, China; [3]State Key Laboratory of Biomacromolecules, Institute of Biophysics, Chinese Academy of Sciences, Beijing, China; [4]The School of Public Health, Xinxiang Medical University, Xinxiang, China; [5]Tsinghua Institute of Multidisciplinary Biomedical Research, Tsinghua University, Beijing, China

*For correspondence:
zhuyun@ibp.ac.cn (YZ);
wangfengchao@nibs.ac.cn (FW)

[†]These authors contributed equally to this work

## eLife Assessment

This **valuable** study reports a critical role of the axonemal protein ANKRD5 in sperm motility and male fertility. **Convincing** data were presented to support the main conclusion. This work will be of interest to biomedical researchers who study ciliogenesis, sperm biology, and male fertility.

**Abstract** Sperm motility is essential for male fertility and depends on the structural integrity of the sperm axoneme, which features a canonical '9 + 2' microtubule arrangement. This structure comprises nine outer doublet microtubules (DMTs) that are associated with various macromolecular complexes. Among them, the nexin–dynein regulatory complex (N-DRC) forms crossbridges between adjacent DMTs, contributing to their stabilization and enabling flagellar bending. In this study, we investigated Ankyrin repeat and EF-hand domain containing 1 (ANKEF1, also known as ANKRD5), a protein highly expressed in the sperm axoneme. We found that ANKEF1 interacts with DRC5/TCTE1 and DRC4/GAS8, two key components of the N-DRC, and these interactions occur independently of calcium regulation. Male *Ankef1*[−/−] mice exhibited impaired sperm motility and infertility. Cryo-electron tomography revealed a typical '9 + 2' axoneme structure with intact DMTs in *Ankef1* null sperm; however, the DMTs showed pronounced morphological variability and increased structural heterogeneity. Notably, ANKEF1 deficiency did not alter ATP levels, reactive oxygen species levels, or mitochondrial membrane potential. These findings suggest that ANKEF1 may attenuate the N-DRC's mechanical buffering—akin to a 'car bumper'—between adjacent DMTs, thereby compromising axonemal stability under high mechanical stress during vigorous flagellar beating.

## Introduction

The interaction between sperm and egg, culminating in embryo formation, is fundamental to sexual reproduction and the continuation of species (*Bhakta et al., 2019*). Male infertility affects approximately 8–12% of the global male population, with defects in sperm motility accounting for over 80% of these cases (*Agarwal et al., 2021*; *Hwang et al., 2021*). Fertilization requires successful spermatogenesis and normal sperm motility (*Lu et al., 2012*). In mammals, sperm acquire motility and fertilizing capacity during transit through the epididymis (*Dey et al., 2019*). This maturation process is essential

for generating functionally competent sperm. Asthenozoospermia, characterized by reduced sperm motility, is a leading cause of clinical infertility; however, its underlying mechanisms remain poorly understood (*Heidary et al., 2020*). Men with poorly motile or immobile sperm are typically infertile unless assisted reproductive techniques (ART), such as gamete intrafallopian transfer, in vitro fertilization (IVF), or intracytoplasmic sperm injection (ICSI), are employed (*Campagne, 2006*). Nevertheless, these ART methods may transmit underlying genetic defects to offspring. Deeper insights into the molecular mechanisms of sperm motility could yield targeted therapies for asthenozoospermia. Rather than bypassing the defect with ICSI, such strategies could directly correct it via modulation of key signaling pathways or gene therapy, potentially offering a cure (*Oscoz-Susino et al., 2025*; *Xu et al., 2018*; *Wang et al., 2021*).

Sperm motility is powered by the rhythmic beating of the flagellar, which is subdivided into the midpiece, principal piece, and endpiece (*Miyata et al., 2020a*). These segments share a conserved core structure—the central axoneme—comprising ~250 proteins that form the main components of the flagellum (*Inaba, 2003*). The axoneme exhibits a characteristic '9 + 2' ultrastructure, featuring nine outer doublet microtubules (DMTs) encircling a central pair of singlet microtubules. Adjacent DMTs are interconnected by the nexin–dynein regulatory complex (N-DRC) (*Yogo, 2022*). The structure and molecular composition of the N-DRC are evolutionarily conserved and central to the regulation of sperm motility (*Awata et al., 2015*; *Bower et al., 2018*; *Bower et al., 2013*).

The N-DRC is a ~1.5 MDa macromolecular complex composed of two primary subdomains: the linker and the base plate (*Bower et al., 2018*; *Bower et al., 2013*; *Heuser et al., 2009*). It also interacts with the outer dynein arms (ODA) via outer–inner dynein linkers, thereby contributing to the regulation of both ODAs and inner dynein arms (IDAs) (*Oda et al., 2013*). Although the N-DRC was initially believed to consist of 11 protein subunits (*Bower et al., 2013*; *Lin et al., 2011*), a twelfth component, CCDC153 (DRC12), was later identified through its interaction with DRC1 (*Ghanaeian et al., 2023*). In situ cryoelectron tomography (cryo-ET) studies in *Chlamydomonas* have elucidated the three-dimensional architecture of the N-DRC, revealing that DRC1, DRC2/CCDC65, and DRC4/GAS8 form the core scaffold (*Oda et al., 2015*). Proteins DRC3/5/6/7/8/11 associate with this core and mediate interactions with other axonemal complexes (*Gui et al., 2019*). Biochemical analyses corroborate these findings and validate the proposed structural model (*Bower et al., 2018*; *Morohoshi et al., 2020*; *Zhang et al., 2021*). Functionally positioned between DMTs, the N-DRC converts microtubule sliding into coordinated axonemal bending by restricting the relative displacement of outer DMTs (*Satir, 1968*; *Summers and Gibbons, 1971*; *Woolley, 1997*). Genetic mutations in N-DRC subunits demonstrate that its structural integrity is crucial for sperm motility. Specifically, mutations in DRC1, DRC2/CCDC65, and DRC4/GAS8 are associated with ciliary motility disorders, leading to primary ciliary dyskinesia (PCD) (*Bower et al., 2018*; *Zhang et al., 2021*). Biallelic truncating mutations in DRC1 induce MMAF in humans, including disassembly of outer DMTs, disorganization of the mitochondrial sheath, and incomplete axonemal assembly (*Zhang et al., 2021*; *Fu et al., 2023*; *Tebbakh et al., 2025*). Similarly, loss of CCDC65 destabilizes the N-DRC, resulting in disorganized axonemes, global microtubule dissociation, and complete asthenozoospermia (*Bower et al., 2018*; *Jreijiri et al., 2024*). Recent mammalian knockout studies further confirmed that loss of DRC2 or DRC4 results in severe sperm flagellar assembly defects, multiple morphological abnormalities of the sperm flagella (MMAF), and complete male infertility, highlighting their indispensable roles in spermatogenesis and reproduction (*Ren et al., 2025*). Homozygous frameshift mutations in DRC3 impair N-DRC assembly and intraflagellar transport, causing severe motility defects despite normal sperm morphology (*Qin et al., 2024*; *Zhou et al., 2023*). In contrast, TCTE1 knockout mice exhibit normal sperm axoneme structure but impaired glycolysis, leading to reduced ATP levels, diminished sperm motility, and male infertility (*Castaneda et al., 2017*). Both *Drc7* and *Iqcg (Drc9)* knockout mice display disrupted '9 + 2' axonemal architecture, complete sperm immotility, and male infertility (*Morohoshi et al., 2020*; *Li et al., 2014*). Although the N-DRC is critical for sperm motility, whether additional regulatory components coordinate its function remains unclear. Here, we demonstrate that ANKEF1 is a novel N-DRC component essential for maintaining sperm motility. Absence of ANKEF1 results in diminished sperm motility and consequent male infertility.

## Results

### *Ankef1* is essential for male fertility

Based on NCBI and single-cell RNA sequencing data, *Ankef1* exhibits testis-specific expression, with particularly high enrichment in the male reproductive system (*Gan et al., 2013*). In mice, ANKEF1 is a protein of 775 amino acids with a molecular weight of 86.9 kDa. Cross-species sequence comparison revealed that ANKEF1 is evolutionarily conserved (*Figure 1—figure supplement 1A*), and alignment via Clustal Omega demonstrated 86% similarity between mouse and human sequences (*Figure 1—figure supplement 1B*). Quantitative PCR confirmed testis-specific expression of *Ankef1*, with no detectable expression in brain, liver, spleen, kidney, ovary, intestine, or stomach (*Figure 1A*). Temporal expression profiling further revealed that *Ankef1* expression begins at postnatal day 21 and reaches a relatively high level around day 35 (*Figure 1B*), coinciding with the onset of sperm maturation.

To investigate its function in vivo, we generated *Ankef1* knockout mice (*Ankef1$^{-/-}$*) on a C57BL/6J background by deleting exons 4–7 using CRISPR/Cas9. Gene deletion was confirmed by PCR using the primers listed in *Supplementary file 1*, *Figure 1C*. Fertility testing revealed that *Ankef1$^{-/-}$* males were capable of mating with wild-type females but failed to produce offspring (*Table 1*), indicating complete male infertility. Since litter size and spermatogenesis were unaffected in *Ankef1$^{+/-}$* males (*Table 1*; *Figure 1D, E*), heterozygotes were used as experimental controls.

Histological analysis revealed no detectable differences in spermatogenesis between *Ankef1$^{-/-}$* and control males (*Figure 1D, E, G*). The testis-to-body weight ratio and overall morphology of the reproductive system were also normal in *Ankef1$^{-/-}$* mice (*Figure 1F*, *Figure 1—figure supplement 1C, D*). Furthermore, hematoxylin and eosin (H&E) staining of testis sections showed that the seminiferous tubules in *Ankef1$^{-/-}$* mice maintained normal architecture and germ cell composition (*Figure 1G*). These findings suggest that male infertility in *Ankef1$^{-/-}$* mice is not a result of defective spermatogenesis, but instead may reflect a functional defect downstream of sperm development.

### Reduced sperm motility in *Ankef1* knockout mice impairs zona pellucida penetration

To determine the cause of infertility in *Ankef1* knockout mice, we performed IVF assays. Control sperm successfully fertilized both cumulus-intact and cumulus-free oocytes (*Figure 2A, B*). In contrast, *Ankef1* null sperm failed to fertilize cumulus-intact oocytes, despite exhibiting normal binding to the zona pellucida (ZP) (*Figure 2A, D*). However, when the ZP was removed, *Ankef1* null sperm were able to fertilize the oocytes and support development to the blastocyst stage (*Figure 2C*). These findings suggest that infertility in *Ankef1* knockout males is primarily due to impaired sperm penetration of the zona pellucida.

Penetration through the cumulus–oocyte complex (COC) requires both functional acrosome reaction and adequate motility (*Kim et al., 2008*). Notably, *Ankef1* null sperm exhibited a normal acrosome reaction when stimulated with the calcium ionophore A23187 (*Figure 2E and F*), indicating that impaired fertilization is not due to a defect in the acrosome reaction.

Since sperm motility plays a vital role in enabling sperm to traverse the cumulus cell layer and the ZP, we next assessed motility using computer-assisted sperm analysis (CASA). Compared with control sperm, *Ankef1* null sperm showed significantly reduced curvilinear velocity (VCL), average path velocity (VAP), and straight-line velocity (VSL) (*Figure 3A*). In addition, parameters reflecting forward progression, including straightness (STR) and linearity (LIN), were also markedly decreased (*Figure 3A*). Based on motility classifications (rapid, medium, slow, and static), *Ankef1* null sperm exhibited a significant decrease in the proportion of rapid sperm and an increase in slow and static subgroups, while the medium subgroup remained unchanged (*Figure 3B*). Both total and progressive motility were significantly impaired in the absence of ANKEF1 (*Figure 3C*). Time-lapse tracking further illustrated the disorganized and limited movement of *Ankef1* null sperm (*Figure 3D* and *Figure 3—videos 1 and 2*), providing visual confirmation of the motility defect. To further assess sperm movement in vivo, we tracked sperm trajectories and analyzed their distribution within the female reproductive tract. Sperm migration analysis performed 6 hr after mating showed a markedly reduced number of *Ankef1* null sperm in the uterus and oviduct (*Figure 3E–G*). These results provide compelling evidence that impaired motility hinders *Ankef1* null sperm from effectively migrating toward and penetrating the egg, thereby contributing to male infertility.

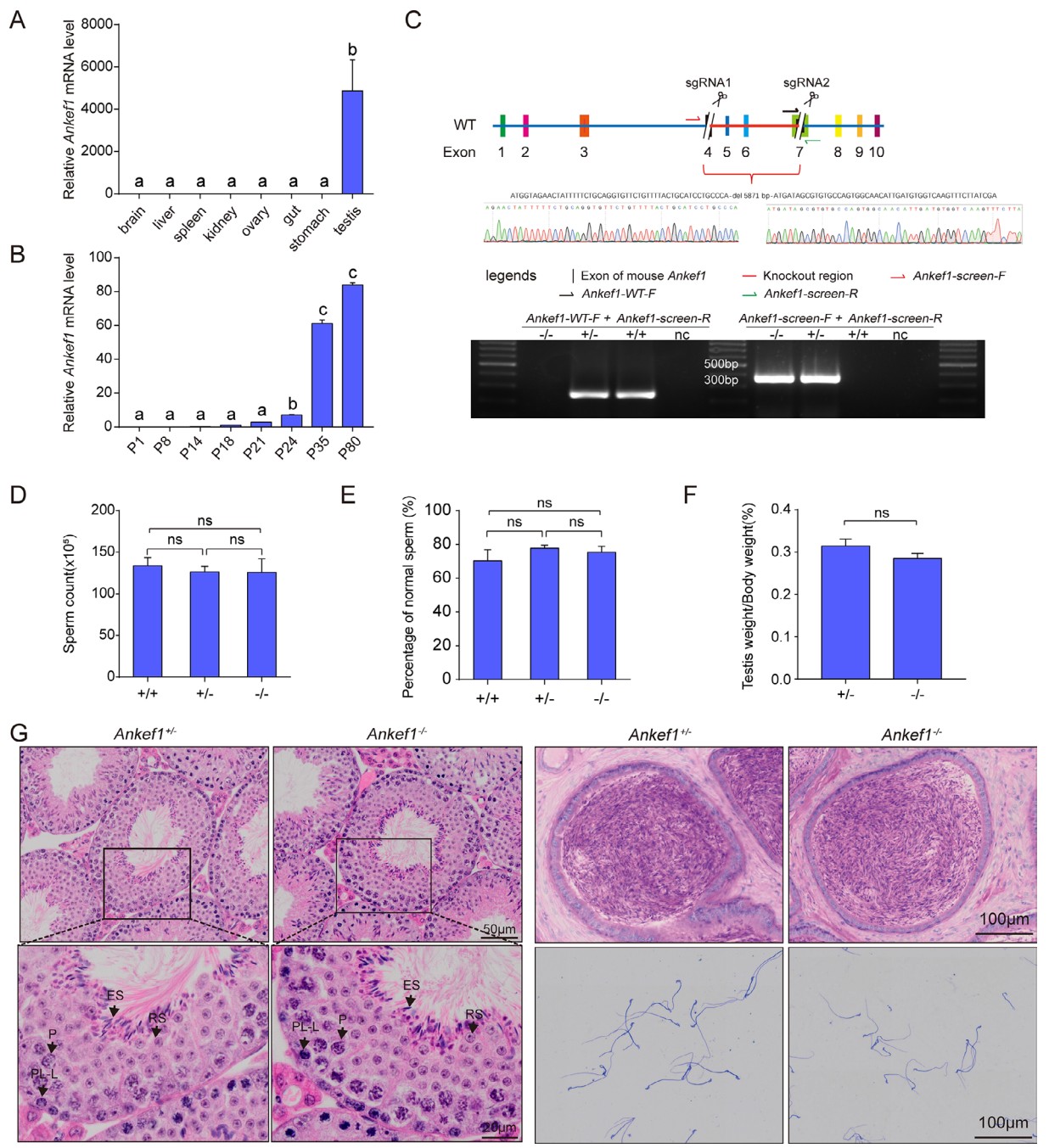

**Figure 1.** *Ankef1* is critical for male reproductive function. (**A**) Relative expression of *Ankef1* mRNA in different tissues of adult mouse. Expression levels were normalized to *Gapdh* mRNA and are presented relative to the gut (set as 1). Different letters above the bars indicate statistically significant differences (p < 0.05). Data are presented as mean ± SEM (*n* = 3 biological replicates). Statistical significance was determined by one-way ANOVA with Dunnett's multiple comparisons test. (**B**) Relative expression of *Ankef1* mRNA in testes at various postnatal days. Expression levels were normalized to *Gapdh* mRNA and are presented relative to the postnatal day 18 (P18, set as 1). Different letters above the bars indicate statistically significant differences (p < 0.05). Data are presented as mean ± SEM (*n* = 3 biological replicates). Statistical significance was determined by one-way ANOVA with Dunnett's multiple comparisons test. (**C**) CRISPR/Cas9 targeting scheme of mouse *Ankef1* and genotyping of *Ankef1* KO mouse. *Ankef1-WT-F+Ankef1-screen-R* (for WT) and *Ankef1-screen-F+Ankef1-screen-R* (for KO). nc, negative control (ddH₂O). (**D, E**) Sperm count and percentage of normal sperm of cauda epididymal. Data are presented as mean ± SEM (*n* = 5 biological replicates). Statistical significance was determined by unpaired two-tailed Student's *t*-test (ns, not significant). (**F**) Testis-to-body weight ratio of adult control and *Ankef1* KO mouse (*n* = 7). Data are presented as mean ± SEM (*n* = 7 biological replicates). Statistical significance was determined by an unpaired two-tailed Student's *t*-test (ns, not significant). (**G**) Hematoxylin and eosin (H&E) staining of mouse testis and epididymis. Coomassie Brilliant Blue R-250 staining of spermatozoa from control and *Ankef1* KO male mouse.

*Figure 1 continued on next page*

*Figure 1 continued*

No significant abnormality was found in *Ankef1* KO male mouse. No overt abnormalities were found in *Ankef1* KO mouse. P, pachytene; ES, elongated sperm; RS, round sperm; SG, spermatogonia; ST, Sertoli cell.

The online version of this article includes the following figure supplement(s) for figure 1:

**Figure supplement 1.** ANKEF1 is conserved between human and other vertebrate model organisms.

## ANKEF1 localizes to the sperm axoneme and tracheal motile cilia

To investigate the biological function of ANKEF1, we generated *Ankef1*-Flag knock-in mice by inserting a Flag tag at the C-terminus via homologous recombination (*Figure 4—figure supplement 1A*). Histological analysis of tissue sections stained with H&E demonstrated that the epididymal structure and cellular composition in *Ankef1*-Flag male mice were comparable to those of wild-type controls (*Figure 4—figure supplement 1B, C*). Furthermore, *Ankef1*-Flag males displayed normal reproductive capacity, indicating that Flag tagging did not disrupt the physiological function of ANKEF1.

Western blot analysis was performed to assess ANKEF1 protein expression across multiple tissues, including liver, lung, kidney, heart, testis, spleen, brain, small intestine, skin, and sperm. ANKEF1 was found to be highly expressed in sperm, with markedly lower levels observed in the testis (*Figure 4A*). To further determine its subcellular localization within sperm, we separated sperm heads and tails via repeated freeze–thaw cycles followed by density gradient centrifugation. Western blot analysis revealed that ANKEF1 was predominantly localized to the sperm tail (*Figure 4B*).

The sperm tail is anatomically divided into three segments: the midpiece, principal piece, and endpiece. Immunofluorescence staining of mature sperm showed that ANKEF1 is primarily localized to the midpiece region (*Figure 4C*). As previously reported, proteins associated with different substructures of the sperm tail can be extracted using detergents of varying solubilizing strengths (*Castaneda et al., 2017*; *Cao et al., 2006*; *Miyata et al., 2021*). To refine the localization of ANKEF1, we employed biochemical fractionation using established markers: BASIGIN (Triton X-100-soluble), Acetylated Tubulin (SDS-soluble), and AKAP4 (SDS-resistant), which, respectively, represent membrane/cytosolic proteins, axonemal proteins, and fibrous sheath/outer dense fiber components (*Castaneda et al., 2017*; *Cao et al., 2006*; *Miyata et al., 2021*). ANKEF1 was primarily enriched in the SDS-soluble fraction, suggesting its association with axonemal structures (*Figure 4D*).

These findings support the hypothesis that the infertility observed in *Ankef1* knockout males may stem from axoneme-related defects in the sperm tail that impair motility. Given that respiratory cilia in mice also possess a canonical axonemal architecture (*Ueno et al., 2012*), we further examined ANKEF1 localization in tracheal tissue (*Figure 4—figure supplement 2*). However, *Ankef1*-deficient mice did not display abnormalities in viability or general locomotor function beyond male infertility, suggesting that ANKEF1 may exert a tissue-specific role, with essential function limited to the male reproductive system.

## ANKEF1 interacts with N-DRC components in the axoneme

To investigate the function of ANKEF1 in the axoneme, we characterized its interactome using liquid chromatography–mass spectrometry (LC–MS) (*Figure 5A*). Among the interacting proteins identified, several known components of the N-DRC were detected, including TCTE1, DRC3/LRRC48, DRC7/CCDC135, and DRC4/GAS8. Notably, TCTE1 ranked among the most abundant hits (*Figure 5A*). Previous studies have shown that *Tcte1* null sperm maintain a normal DMT structure but exhibit reduced motility (*Castaneda et al., 2017*), a phenotype also observed in *Ankef1* null sperm. Although ANKEF1 was identified in the TCTE1 LC–MS profile, a direct interaction between these proteins had

**Table 1.** Knockout of *Ankef1* causes male infertility in mice.

| | Females | Plugs | Litters | Mean litter size |
|---|---|---|---|---|
| *WT* (*n* = 6) | 12 | 11 | 10 | 8 |
| *Ankef1⁺/⁻* (*n* = 6) | 12 | 12 | 10 | 8.5 |
| *Ankef1⁻/⁻* (*n* = 6) | 12 | 10 | 0 | 0 |

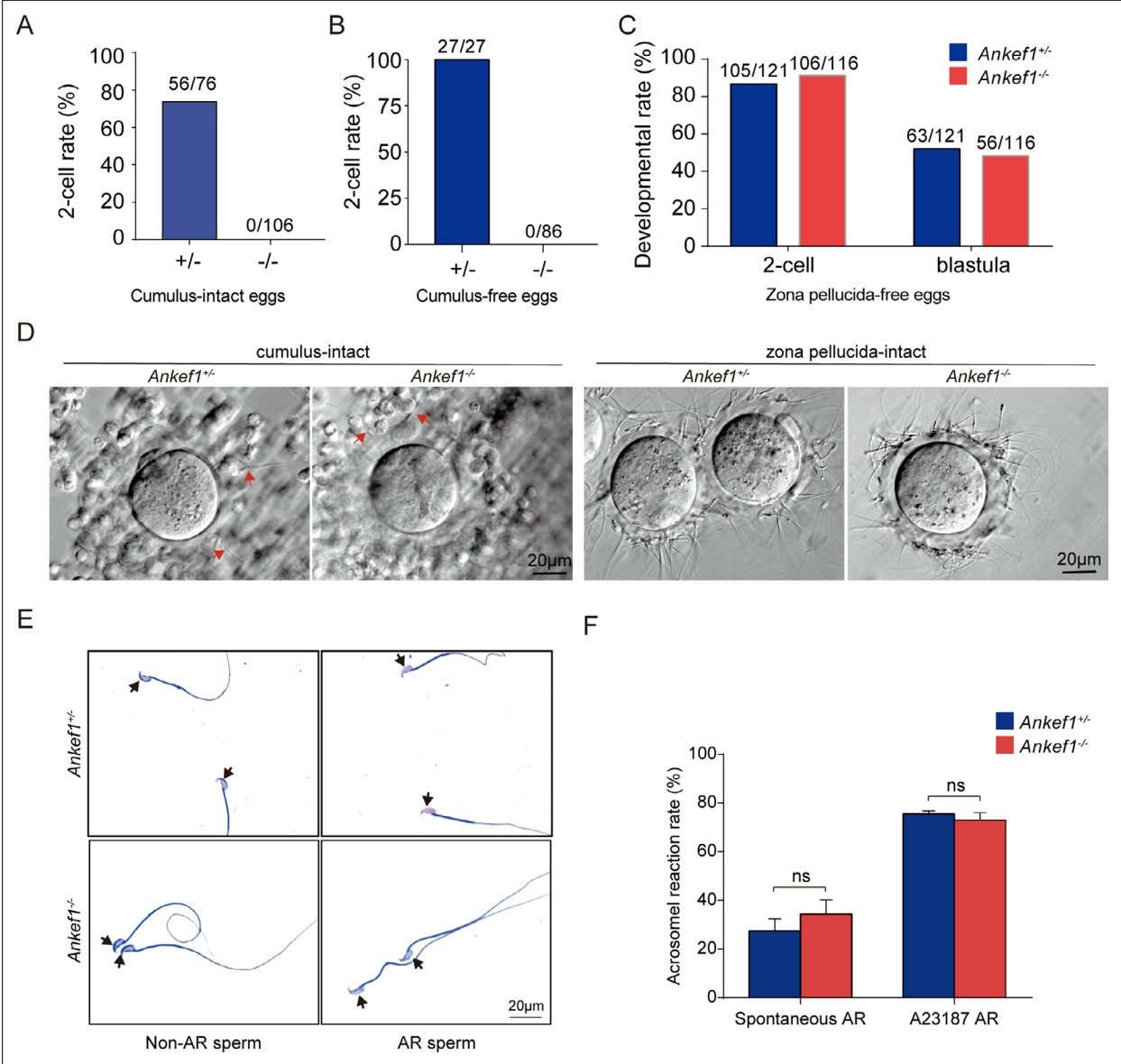

**Figure 2.** Evaluation of in vitro fertilization (IVF) capacity of *Ankef1* KO sperm. (**A–C**) Fertilization rate of IVF using control and *Ankef1* KO spermatozoa. Three types of oocytes (cumulus-intact, cumulus-free, and zona pellucida-free) were used for IVF. (**D**) Egg observation after IVF. After 4 hr of incubation, both the control and *Ankef1* KO sperm could penetrate cumulus oophorus as indicated by the red arrow and have the ability to bind to the zona pellucida. (**E, F**) Sperm were incubated in capacitation medium treated with A23187 (dissolved in DMSO) and DMSO (dissolvent control group) and stained with Coomassie Brilliant Blue R-250. A black arrow indicates the intact or disappeared acrosome. Data are mean ± SEM. Sample sizes (biological replicates) were *Ankef1*$^{+/-}$ mice, $n = 4$ per group; *Ankef1*$^{-/-}$ mice, $n = 3$ per group. Within each genotype, the DMSO and A23187 groups were compared by an unpaired two-tailed *t*-test (ns, not significant).

not been previously validated (*Castaneda et al., 2017*). It is possible that the interactions of ANKEF1 with DRC3 and DRC7 are indirect rather than direct physical bindings.

To confirm these interactions, we performed co-immunoprecipitation assays in HEK293T cells, testing interactions between ANKEF1 and 11 DRC components (excluding DRC6, which failed to express) (*Figure 5B, C* and *Figure 5—figure supplement 1A*). These experiments confirmed that ANKEF1 interacts specifically with TCTE1 and DRC4/GAS8, but not with other tested DRC components (*Figure 5B, C* and *Figure 5—figure supplement 1A, B*).

Structurally, ANKEF1 contains two ANK repeat domains and one EF-hand domain. The ANK domain is a well-known scaffold for protein–protein interactions, while the EF-hand domain typically functions as a calcium-binding motif, though approximately one-third of EF-hand domains lack calcium-binding

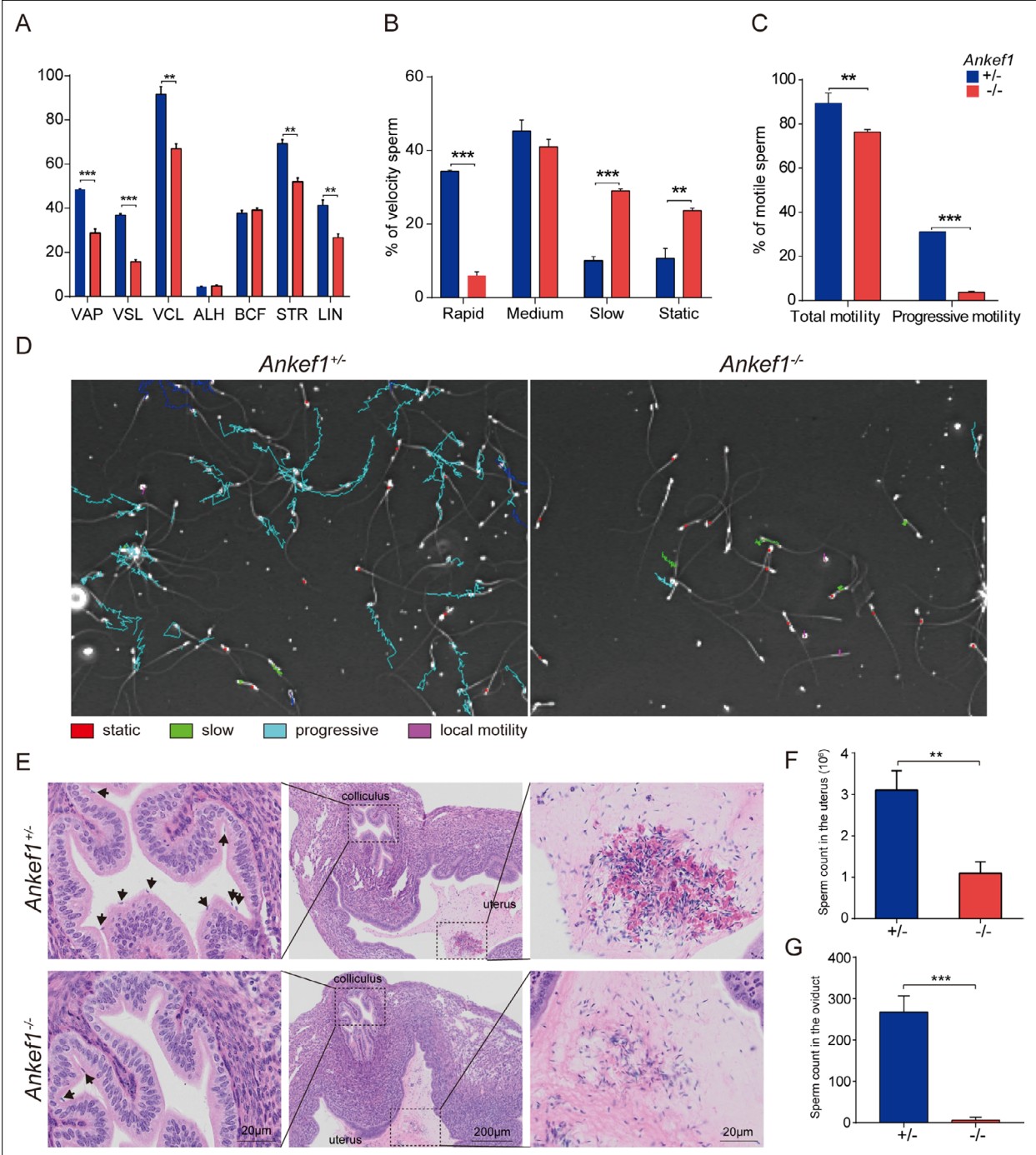

**Figure 3.** Sperm motility of *Ankef1* KO male mouse. (**A**) Average path velocity (VAP), straight-line velocity (VSL), curvilinear velocity (VCL), amplitude of lateral head displacement (ALH), beat cross frequency (BCF), straightness (STR), and linearity (LIN) of sperm from control and *Ankef1* KO mouse. (**B**) Proportions of sperm at different velocity levels in control and *Ankef1* KO mouse. (**C**) Knockout mouse had lower motile sperm (total motor capacity) and progressive motile sperm (progressive motor capacity) than control. (**D**) Trajectories of sperm per second. The meanings of different colors are shown in the graph. (**E**) Impaired migration of *Ankef1* KO sperm from uterus into oviducts. The black arrow indicates sperm. (**F**) Uterine sperm counts after mating. (**G**) Oviduct sperm counts after mating. For both panels, data (mean ± SEM) compare control and *Ankef1* KO mice, analyzed by an unpaired two-tailed *t*-test (uterus, *n* = 6 biological replicates, **p < 0.01; oviducts, *n* = 3 biological replicates, ***p < 0.001).

The online version of this article includes the following video(s) for figure 3:

**Figure 3—video 1.** Sperm from *Ankef1*+/− mouse.

https://elifesciences.org/articles/105321/figures#fig3video1

*Figure 3 continued on next page*

*Figure 3 continued*

**Figure 3—video 2.** Sperm from *Ankef1⁻ᐟ⁻* mouse.

https://elifesciences.org/articles/105321/figures#fig3video2

ability (*Lewit-Bentley and Réty, 2000*). Protein truncation analysis revealed that the ANK2 domain is necessary for interaction with TCTE1 (*Figure 5D, E*), whereas binding to DRC4/GAS8 requires both ANK1 and ANK2 domains (*Figure 5F*). The EF-hand domain was not essential for interaction with either protein. Furthermore, calcium supplementation did not alter the binding of ANKEF1 to DRC4/GAS8 or TCTE1, as shown by western blot analysis (*Figure 5G*), indicating that these interactions are calcium-independent.

Additional candidate interactors identified in the LC–MS dataset include KRT77, a cytoskeletal protein known to maintain structural stability. It may contribute to reinforcing the physical linkage between the N-DRC and adjacent DMTs through interaction with ANKEF1. Recent structural studies have positioned ANKEF1 within the distal lobe of the N-DRC, where its positively charged surface may facilitate electrostatic interactions with glutamylated tubulin on neighboring DMTs (*Leung et al., 2025*). KRT77 may further modulate this interaction via post-translational modifications such as phosphorylation, thereby enhancing the structural integrity of the flagellum under mechanical stress during sperm motility.

Other candidate interactors include members of the Rab GTPase family, which are implicated in intraflagellar transport and membrane trafficking. RAB2A, for example, may regulate the targeted delivery of ANKEF1 or other N-DRC components to assembly sites within the axoneme through vesicle-mediated transport. Its GTPase activity may also serve as a regulatory node linking signaling pathways to axonemal remodeling. However, given the complexity of mass spectrometry datasets, we cannot exclude the possibility that some observed interactions are false positives arising from nonspecific factors such as electrostatic interactions, detergent-mediated membrane disruption, protein aggregation, or high-abundance protein interference.

Sperm motility is highly dependent on energy metabolism. The mitochondrial sheath, located in the sperm tail midpiece, is responsible for ATP production. Previous studies have shown that *Tcte1* null sperm exhibit reduced ATP levels (*Castaneda et al., 2017*). However, in contrast, *Ankef1* KO sperm displayed no significant change in ATP content (*Figure 6E*), suggesting that ANKEF1 functions independently of ATP synthesis.

The sperm midpiece is also a major site of reactive oxygen species (ROS) generation (*Koppers et al., 2008*; *Kothari et al., 2010*). Proper regulation of ROS is essential for key sperm functions, including motility, capacitation, acrosome reaction, fertilization, and hyperactivation (*Amaral et al., 2013*, *Baumber et al., 2003*; *Gibb et al., 2014*). Mitochondrial membrane potential (MMP) is correlated with sperm motility (*Barbagallo et al., 2020*), decreased MMP (depolarization) indicates mitochondrial dysfunction, while increased MMP (hyperpolarization) leads to excess ROS. We assessed MMP in spermatozoa using tetramethylrhodamine methyl ester (TMRM) staining and fluorescence imaging (*Figure 6—figure supplement 1A, B*) and measured ROS levels using DCFH-DA staining followed by fluorescence imaging (*Figure 6—figure supplement 1C, D*). No significant differences were observed between *Ankef1* null sperm and controls, indicating that ANKEF1 deficiency does not significantly affect sperm ATP levels or mitochondrial function.

## The ANKEF1-KO sperm axoneme retains the canonical '9 + 2' architecture

Loss of certain DRC components has been shown to affect the expression of other N-DRC subunits. For example, expression of DRC2/3/4 is reduced in *Drc1* mutant mice (*Zhang et al., 2021*), while DRC1/3/5/11 expression is downregulated in *Chlamydomonas Drc2* mutants (*Zhang et al., 2021*). To evaluate the impact of ANKEF1 deletion on sperm protein composition, we performed quantitative proteomic profiling using 4D-SmartDIA. Among the 4880 quantifiable proteins detected by mass spectrometry, 126 were differentially expressed by more than 1.5-fold, with 10 upregulated and 116 downregulated proteins (*Figure 6A, B*).

Notably, none of the annotated N-DRC components were included in the list of significantly differentially expressed proteins. To further examine expression changes of specific N-DRC components, their intensity values were extracted, normalized, and subjected to *t*-test analysis. Due to technical

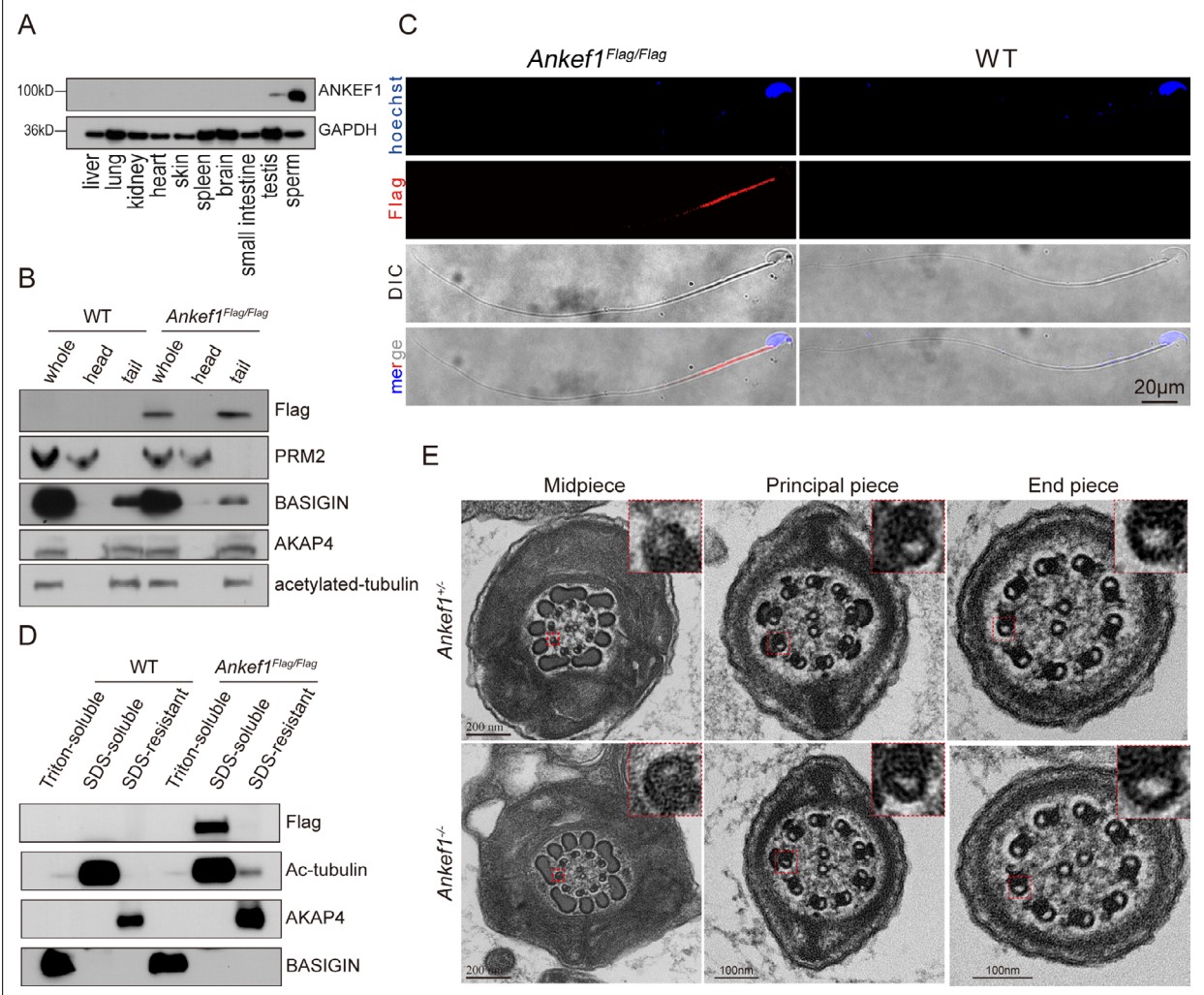

**Figure 4.** ANKEF1 is located in the midpiece of sperm axoneme. (**A**) Immunoblotting analysis of various mouse tissues. GAPDH was used as the loading control. (**B**) Head and tail separation of mouse spermatozoa. ANKEF1-Flag was detected in the tail fraction. PRM2 was used as a marker for sperm head. BASIGIN, AKAP4, and acetylated tubulin were detected as a marker for tails. (**C**) Immunofluorescence staining results of spermatozoa from wild-type and ANKEF1-Flag mouse using anti-Flag antibody (red: anti-Flag signal; Hoechst: blue). (**D**) Fractionation of sperm proteins using different lysis buffers. ANKEF1-Flag was found in the SDS-soluble fraction that contains axonemal proteins. BASIGIN, acetylated tubulin, and AKAP4 were detected as a marker for Triton-soluble, SDS-soluble, and SDS-resistant fractions, respectively. (**E**) Transmission electron microscopy (TEM) of sperm tails from control and *Ankef1* KO mice. Cross-sections of the midpiece, principal piece, and end piece were examined. Red dashed boxes highlight regions of interest, and the magnified views of these boxed areas are shown in the upper right corner of each image. In three independent experiments, 20 sperm cross-sections per mouse were analyzed for each group, with consistent results observed.

The online version of this article includes the following source data and figure supplement(s) for figure 4:

**Source data 1.** PDF file containing original western blots for *Figure 4A* with relevant bands.

**Source data 2.** Original files for western blot analysis displayed in *Figure 4A*.

**Source data 3.** PDF file containing original western blots for *Figure 4B* with relevant bands.

**Source data 4.** Original files for western blot analysis displayed in *Figure 4B*.

**Source data 5.** PDF file containing original western blots for *Figure 4D* with relevant bands.

**Source data 6.** Original files for western blot analysis displayed in *Figure 4D*.

**Figure supplement 1.** Generation and fertility assessment of *Ankef1*-Flag mice.

**Figure supplement 2.** ANKEF1 was expressed in mouse respiratory cilia.

**Figure supplement 3.** The deformed doublet microtubule (DMT) in the transmission electron microscopy (TEM) results.

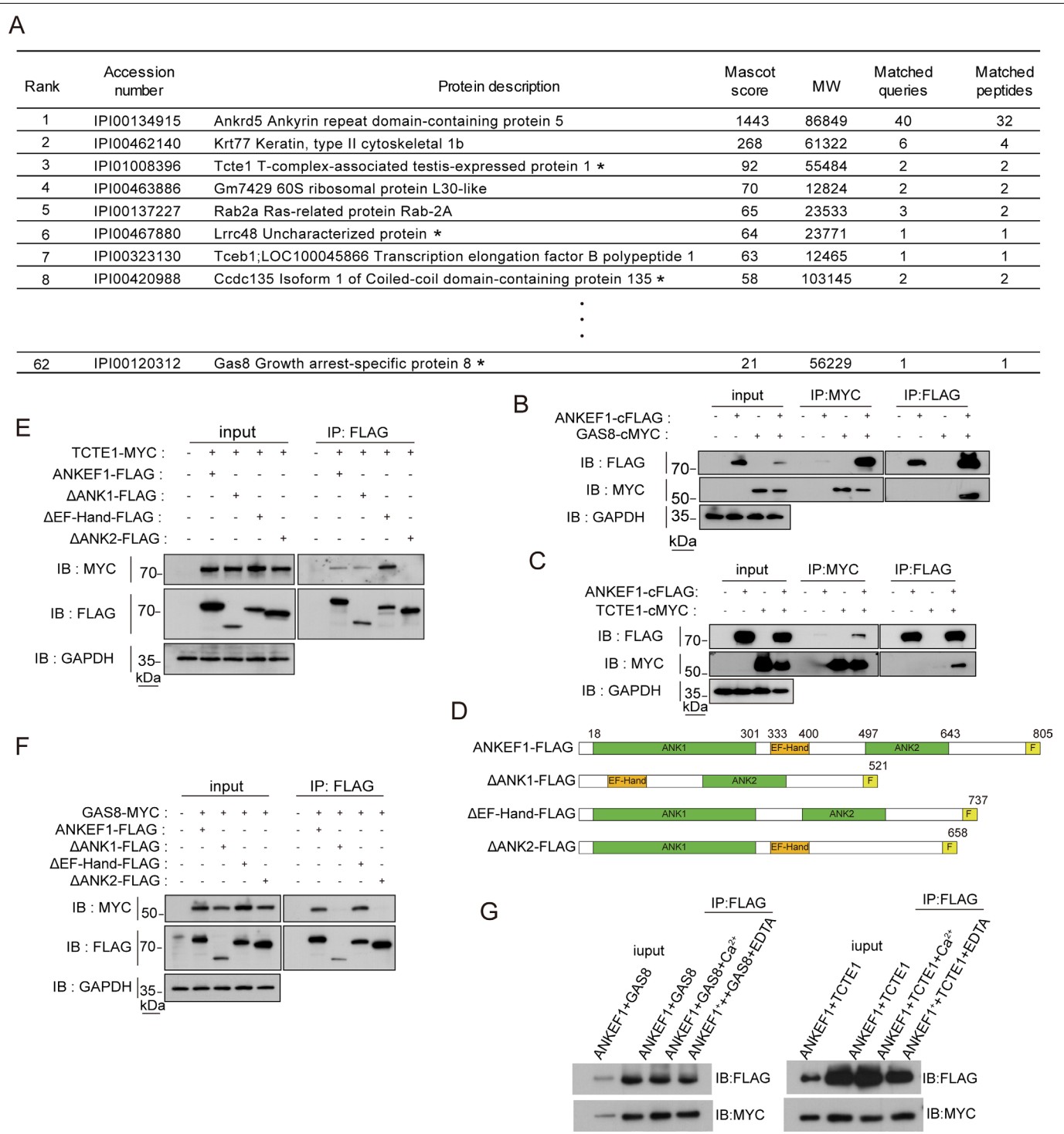

**Figure 5.** ANKEF1 is a component of nexin–dynein regulatory complex (N-DRC) in sperm flagella. (**A**) Identification of sperm proteins in liquid chromatography–mass spectrometry (LC–MS/MS) analysis. Black star indicates N-DRC components. (**B, C**) Individual DRC components were coexpressed in HEK293T cells. Immunoprecipitation of ANKEF1-Flag resulted in the co-precipitation of GAS8-MYC and TCTE1-MYC. Similarly, immunoprecipitation of GAS8-MYC and TCTE1-MYC also led to the co-precipitation of ANKEF1-Flag. (**D**) Schematic of various truncated ANKEF1 vectors. Flag-tag was linked posterior to the C-terminal of ANKEF1. Green and yellow boxes show the ANK domain and EF-hand domain of ANKEF1, respectively. Light yellow boxes indicate Flag tag. (**E, F**) The interaction between various truncated ANKEF1-Flag and TCTE1-MYC or GAS8-MYC was confirmed by co-IP followed by WB analysis using anti-Flag and anti-MYC antibodies. (**G**) Effect of calcium ion and EDTA treatment on the interaction of ANKEF1 with GAS8 and TCTE1.

*Figure 5 continued on next page*

*Figure 5 continued*

The online version of this article includes the following source data and figure supplement(s) for figure 5:

**Source data 1.** PDF file containing original western blots for *Figure 5B* with relevant bands and treatments.

**Source data 2.** Original files for western blot analysis displayed in *Figure 5B*.

**Source data 3.** PDF file containing original western blots for *Figure 5C* with relevant bands and treatments.

**Source data 4.** Original files for western blot analysis displayed in *Figure 5C*.

**Source data 5.** PDF file containing original western blots for *Figure 5E* with relevant bands and treatments.

**Source data 6.** Original files for western blot analysis displayed in *Figure 5E*.

**Source data 7.** PDF file containing original western blots for *Figure 5F* with relevant bands and treatments.

**Source data 8.** Original files for western blot analysis displayed in *Figure 5F*.

**Source data 9.** PDF file containing original western blots for *Figure 5G* with relevant bands and treatments.

**Source data 10.** Original files for western blot analysis displayed in *Figure 5G*.

**Source data 11.** Original files for *Figure 5A*.

**Figure supplement 1.** Identify the interaction of ANKEF1 and other nexin–dynein regulatory complex (N-DRC) components.

**Figure supplement 1—source data 1.** PDF files containing the original western blots, indicating relevant bands, for the co-immunoprecipitation of ANKEF1-Flag with DRC1.

**Figure supplement 1—source data 2.** Original file of Western blot analysis of co-immunoprecipitation of ANKEF1-Flag and DRC1.

**Figure supplement 1—source data 3.** PDF files containing the original western blots, indicating relevant bands, for the co-immunoprecipitation of ANKEF1-Flag with DRC2.

**Figure supplement 1—source data 4.** Original file of Western blot analysis of co-immunoprecipitation of ANKEF1-Flag and DRC2.

**Figure supplement 1—source data 5.** PDF files containing the original western blots, indicating relevant bands, for the co-immunoprecipitation of ANKEF1-Flag with LRRC48.

**Figure supplement 1—source data 6.** Original file of Western blot analysis of co-immunoprecipitation of ANKEF1-Flag and LRRC48.

**Figure supplement 1—source data 7.** PDF files containing the original western blots, indicating relevant bands, for the co-immunoprecipitation of ANKEF1-Flag with EFCAB2.

**Figure supplement 1—source data 8.** Original file of Western blot analysis of co-immunoprecipitation of ANKEF1-Flag and EFCAB2.

**Figure supplement 1—source data 9.** PDF files containing the original western blots, indicating relevant bands, for the co-immunoprecipitation of ANKEF1-Flag with IQCG.

**Figure supplement 1—source data 10.** Original file of Western blot analysis of co-immunoprecipitation of ANKEF1-Flag and IQCG.

**Figure supplement 1—source data 11.** PDF files containing the original western blots, indicating relevant bands, for the co-immunoprecipitation of ANKEF1-Flag with IQCD.

**Figure supplement 1—source data 12.** Original file of Western blot analysis of co-immunoprecipitation of ANKEF1-Flag and IQCD.

**Figure supplement 1—source data 13.** PDF files containing the original western blots, indicating relevant bands, for the co-immunoprecipitation of ANKEF1-Flag with DRC7.

**Figure supplement 1—source data 14.** Original file of Western blot analysis of co-immunoprecipitation of ANKEF1-Flag and DRC7.

**Figure supplement 1—source data 15.** PDF files containing the original western blots, indicating relevant bands, for the co-immunoprecipitation of ANKEF1-Flag with IQCA1.

**Figure supplement 1—source data 16.** Original file of Western blot analysis of co-immunoprecipitation of ANKEF1-Flag and IQCA1.

limitations, DRC6 and DRC12 were not detected. However, DRC11 (IQCA1) displayed a statistically significant change in expression, although validation by western blot was not possible due to the lack of a commercial antibody (*Figure 6C*).

Given the reported link between TCTE1 deficiency, glycolytic defects, and reduced sperm motility through lowered ATP levels (*Castaneda et al., 2017*), we conducted Gene Set Enrichment Analysis (GSEA) focusing on glycolysis-related pathways. The analysis yielded no significant enrichment, which aligns with our direct ATP quantification results, indicating that *Ankef1* knockout does not impair ATP production in sperm (*Figure 6D, E*). To further evaluate structural integrity of key axonemal elements, including RS, IDA, and ODA, immunofluorescence staining was performed on spermatozoa from *Ankef1*⁻/⁻ and control mice. No discernible differences were observed (*Figure 6—figure supplement 2*).

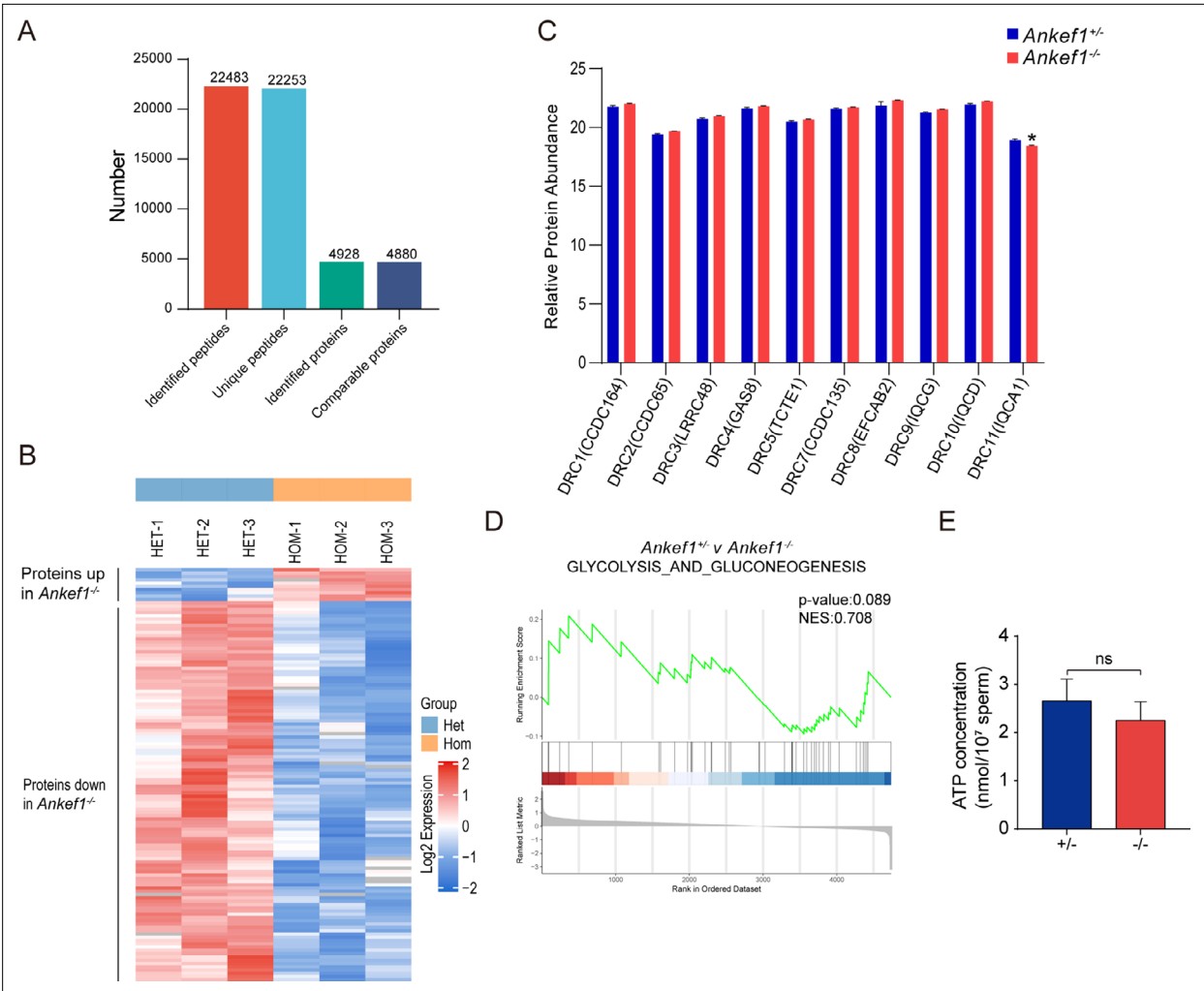

**Figure 6.** Absence of ANKEF1 does not affect energy metabolism. (**A**) The differentially expressed proteins of *Ankef1*[+/–] and *Ankef1*[–/–] were identified by 4D-SmartDIA. (**B**) Heatmap of relative protein abundance changes between control and knockout mouse sperm. (**C**) Differences in the expression of nexin–dynein regulatory complex (N-DRC) protein components identified by mass spectra. (**D**) Gene Set Enrichment Analysis (GSEA) of glycolysis and gluconeogenesis. (**E**) Measured levels of ATP between wild-type and *Ankef1* null sperm. Data are presented as mean ± SEM (*n* = 3 biological replicates). Statistical significance was determined by an unpaired two-tailed Student's *t*-test (*p < 0.05, ns, not significant).

The online version of this article includes the following source data and figure supplement(s) for figure 6:

**Source data 1.** Source data files for panels B and D of *Figure 6* provide the whole-sperm proteomic profiling results from ANKEF1-knockout and control mice.

**Figure supplement 1.** The mitochondrial membrane potential and reactive oxygen species (ROS) levels of *Ankef1* null sperm were normal.

**Figure supplement 2.** Immunofluorescence results of ANKEF1-null sperm and control.

The sperm axoneme is organized in a characteristic '9 + 2' arrangement, consisting of nine outer DMTs surrounding a central pair of singlet microtubules. Structural defects in this 9 + 2 arrangement may impair motility, but TEM of sperm flagella did not reveal any structural defects in the overall '9 + 2' structure of the *Ankef1*[–/–] mice axonemes (*Figure 4E*).

To further investigate the potential effects of *Ankef1* deficiency on axonemal architecture, we collected approximately 160 tilt series of *Ankef1* null sperm axoneme using cryo-focused ion beam (cryo-FIB) milling followed by cryo-ET (*Figure 7—figure supplement 1*). Data pre-processing and reconstruction were performed by Warp and AreTOMO (*Kremer et al., 1996*; *Tegunov and Cramer, 2019*; *Zheng et al., 2022*; *Figure 7—figure supplement 2*). In the original tomograms obtained, we found that the overall '9 + 2' structure of sperm axonemes remained intact regardless of side view or top view, and there was no significant difference from that of WT mouse sperm axonemes (*Figure 7A*;

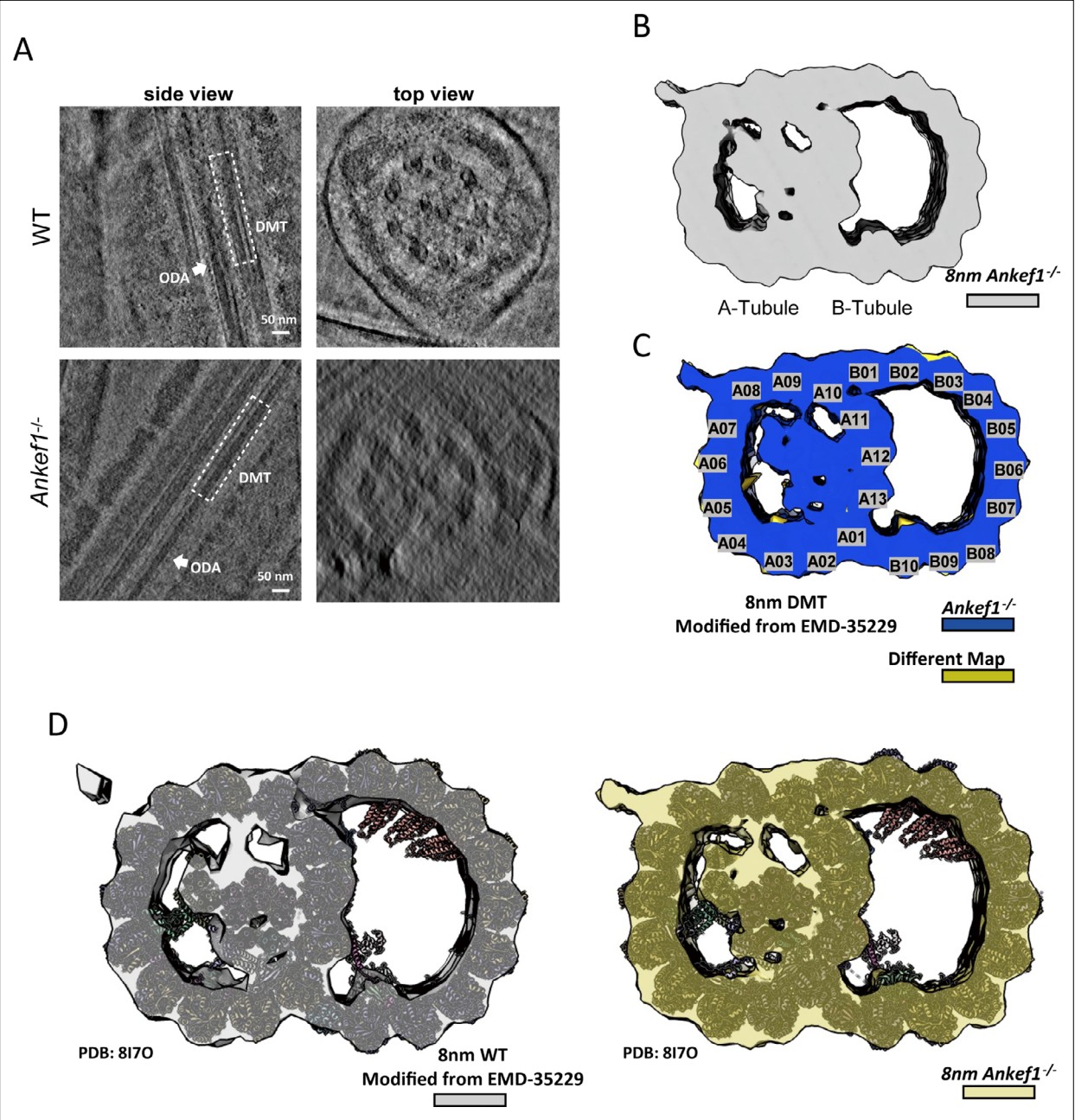

**Figure 7.** The overall structure of *Ankef1*-KO mouse sperm doublet microtubule (DMT). (**A**) Side view and top sectional view of WT/*Ankef1*⁻/⁻ mouse sperm axoneme are shown in the tomogram slices. DMT and outer dynein arm (ODA) are marked with white dashed lines and white arrows, respectively. (**B**) The cryo-EM map of *Ankef1*⁻/⁻ mouse sperm DMT with an 8 nm repeat was obtained by sub-tomogram analysis. (**C**) Loss of density in *Ankef1*⁻/⁻ DMT structure. The transverse sectional view of DMT is shown. The lost density (khaki color) was obtained by subtracting the density map of *Ankef1*⁻/⁻ DMT from that of the WT DMT. (**D**) The model of 16 nm-repeats WT DMT (PDB: 8I7O) was fitted in the 8 nm repeat WT DMT map and *Ankef1*⁻/⁻ DMT map. The 8 nm repeats DMT density map was obtained by summing two 16 nm repeats DMTs that were staggered 8 nm apart from each other.

The online version of this article includes the following figure supplement(s) for figure 7:

**Figure supplement 1.** Cryo-focused ion beam (cryo-FIB) milling and the half-map Fourier Shell Correlation (FSC) of the *Ankef1*⁻/⁻ mouse sperm axoneme.

**Figure supplement 2.** The data processing of ANKEF1-KO mouse sperm doublet microtubule (DMT).

**Figure supplement 3.** The states of doublet microtubule (DMT) particles in sperm of *Ankef1*-KO mouse.

*Tai et al., 2023*), which was similar to TEM results. In addition, the presence of DMT accessory structures RS and ODA was observed in both WT and ANKEF1-KO tomograms (*Figure 7A*, *Figure 7— figure supplement 3*).

Among the 89 tomograms with good quality, we carried out manual selection of DMT particles and applied RELION, Dynamo, and Warp/M to carry out sub-tomogram averaging (STA) analysis (*Scheres, 2012*; *Tegunov et al., 2021*; *Zivanov et al., 2018*). The 8 nm repeat DMT structure of ANKEF1-KO sperm axoneme with a resolution of 24 Å was obtained (*Figure 7—figure supplements 1B and 2*), and the overall structure of its A- and B-tubes was complete (*Figure 7B, D*). We compared the DMT density map of ANKEF1-KO mouse sperm with that of WT mouse sperm (modified from EMD-35229) and found that there was no significant difference between them, except for slight variations in density near A05 of tube A and B10 of tube B (*Figure 7C, D*; *Tai et al., 2023*). The known mouse sperm DMT model was fitted into two density maps, and all 16-nm repeat MIPs including tubulins could be well fitted. This suggests that the loss of ANKEF1 has no significant effect on the overall structure of axoneme and the component proteins of DMT.

## ANKEF1 depletion may impair the buffering function between adjacent DMTs in the axoneme

During particle picking of DMT fibers, we observed notable differences in the transverse sections of axonemal DMT particles between ANKEF1-KO and WT sperm. While both A- and B-tubes were discernible in both groups, DMTs in ANKEF1-KO sperm exhibited a markedly more irregular morphology. In WT sperm, DMTs typically appeared circular in cross-section, whereas in ANKEF1-KO sperm, they frequently adopted a polygonal or extruded appearance (*Figure 7—figure supplement 3B, D*). Notably, some ANKEF1-KO DMTs appeared partially open at the A- and B-tube junctions (*Figure 7—figure supplement 3B, D*, *Figure 8*).

During the STA process, many ANKEF1-KO particles were either misaligned or displayed significant structural deformation, particularly affecting the B-tube (*Figure 9*, *Figure 7—figure supplement 3E*). Upon re-examining the TEM data in light of the cryo-ET findings, similar abnormalities were observed in the TEM images (*Figure 4E*, *Figure 4—figure supplement 3B*). Notably, both intact and deformed DMT structures were consistently observed in both TEM and STA analyses, with the deformation of the B-tube being more obvious (*Figure 4E*, *Figure 4—figure supplement 3*). During the STA process, we could retain only ~10% of the DMT particles to obtain the final density map for ANKEF1-KO sperm (*Figure 7—figure supplement 3E*), compared to ~70% in the WT dataset as previously reported (*Tai et al., 2023*). The resulting density map from ANKEF1-KO sperm also exhibited peripheral roughness, indicative of substantial structural heterogeneity (*Figure 7—figure supplement 3E*). Even after excluding a large proportion of deformed particles, the final averaged map still presented noticeable artifacts, suggesting that although the overall architecture of the DMTs is preserved, its structural integrity is significantly compromised (*Figure 7—figure supplement 3E*). Furthermore, attempts to resolve the 96 nm repeat structure did not yield clear densities for radial spokes (RSs) (*Figure 7—figure supplement 3F*), suggesting that ANKEF1 deficiency may also impair the stability of accessory structures, such as RSs (*Satir, 1968*; *Summers and Gibbons, 1971*; *Woolley, 1997*). In the raw tomograms, RSs in ANKEF1-KO sperm appeared more irregularly arranged compared to those in WT controls (*Figure 7—figure supplement 3A, C*).

Following the submission of this work, ANKEF1 was reported to localize at the head of the N-DRC, interacting simultaneously with DRC11, DRC7, DRC4, and DRC5 (*Leung et al., 2025*). These findings are consistent with our in vitro data demonstrating interactions between ANKEF1 and both DRC4 and DRC5 (*Figure 8C–F*). However, the aforementioned study utilized isolated and purified DMT preparations, leaving the precise spatial relationship between ANKEF1 and neighboring DMTs unresolved. To address this, we fitted the published structure of ANKEF1 (PDB entry: 9FQR) into the in situ 96 nm DMT repeat map from mouse sperm (EMD-27444), revealing that ANKEF1 resides approximately 3 nm from the adjacent DMT (*Figure 8G*). The N-DRC is often analogized to a 'car bumper', serving as a mechanical buffer between adjacent DMTs during vigorous axonemal motion. Given the pronounced DMT deformation observed in our cryo-ET data (*Figure 7—figure supplement 3E*), we propose that ANKEF1 contributes to this buffering function. Loss of ANKEF1 may compromise the structural resilience of the N-DRC, thereby diminishing its ability to protect adjacent DMTs from mechanical stress

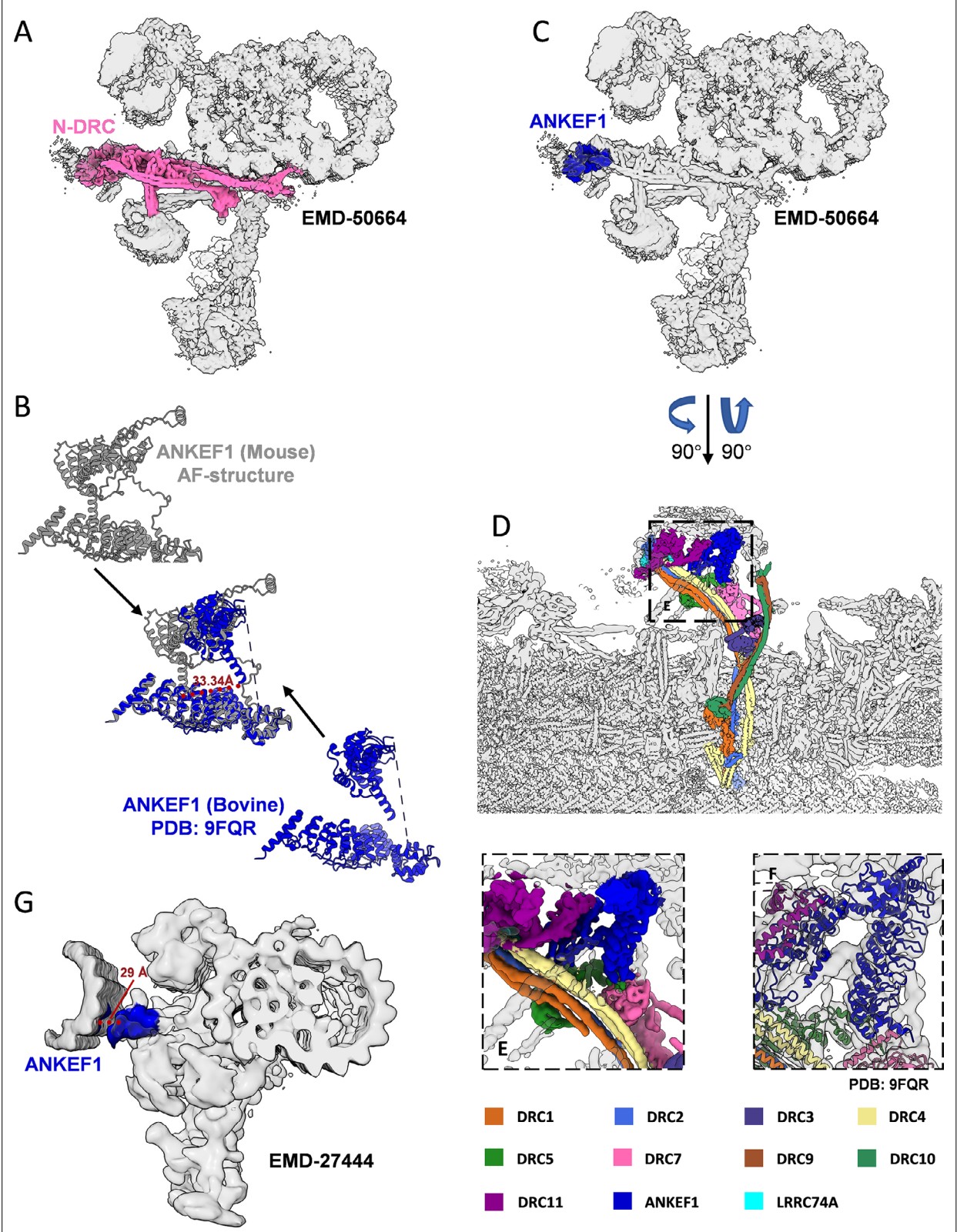

**Figure 8.** Localization of ANKEF1 in sperm axoneme. (**A**) Localization of nexin–dynein regulatory complex (N-DRC) in 96 nm repeats doublet microtubule (DMT) of mammalian sperm (EMD-50664). (**B**) Comparison of AlphaFold predicted mouse ANKEF1 structures with known bovine ANKEF1 structures (PDB: 9FQR). (**C**) Localization of ANKEF1 in 96 nm repeats DMT map (EMD-50664). (**D**) Structure and composition of N-DRC in 96 nm repeats DMT of mammalian sperm (EMD-50664). (**E**) The relationship between ANKEF1 and its interacting proteins as shown in the electron microscope density

*Figure 8 continued on next page*

*Figure 8 continued*

map (EMD-50664). (**F**) The relationship between ANKEF1 and its interacting proteins as shown in 96 nm repeats DMT model of mammalian sperm (PDB: 9FQR). (**G**) Localization of ANKEF1 in the in situ mouse sperm 96 nm repeats DMT map (EMD-27444).

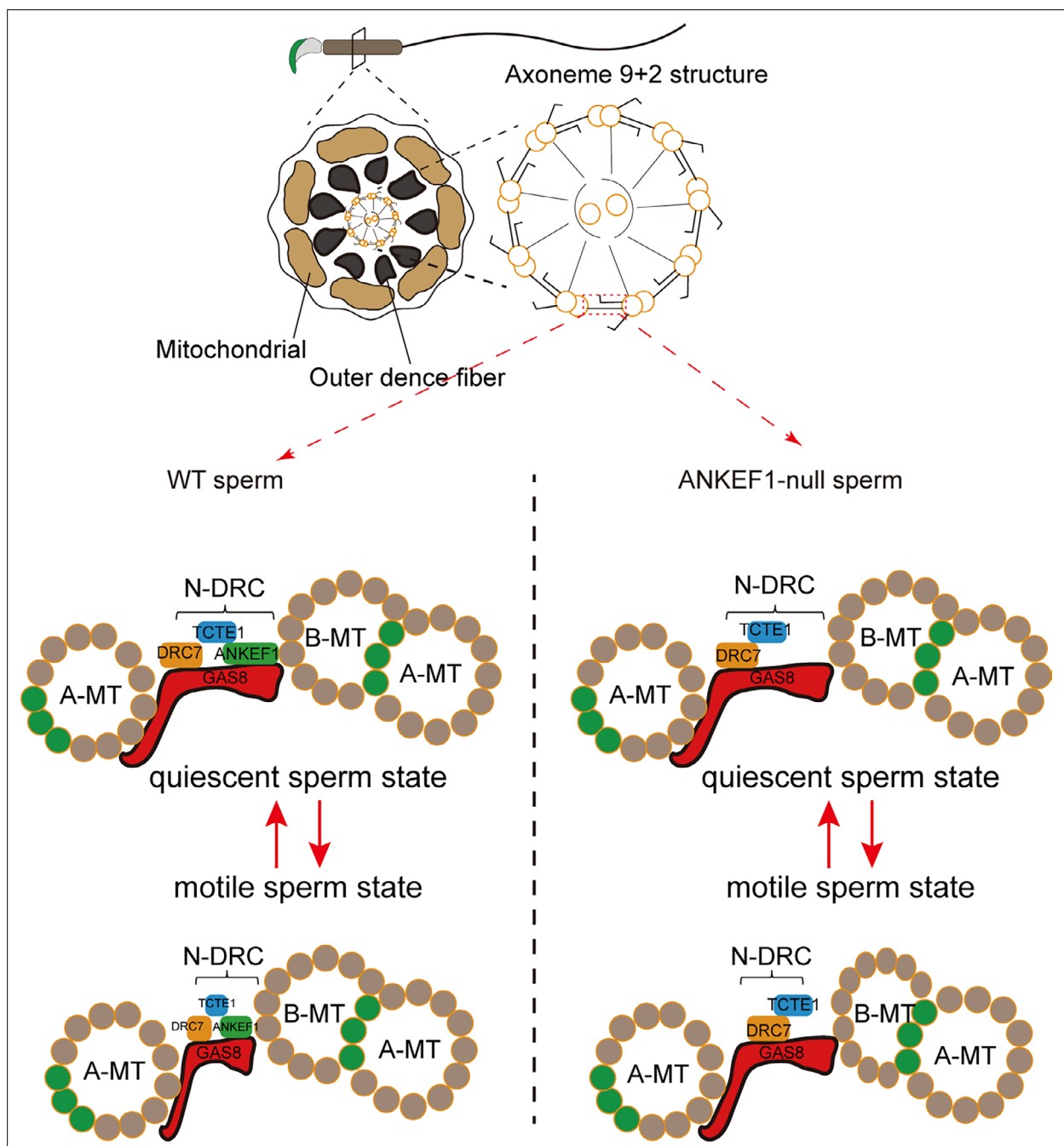

**Figure 9.** The functional model of ANKEF1. ANKEF1, as a component of the nexin–dynein regulatory complex (N-DRC) connecting adjacent doublet microtubules (DMTs), enhances the elasticity of the N-DRC. During sperm movement, the flagellar beating causes adjacent DMT to move toward each other, and the intact N-DRC structure provides good cushioning. In ANKEF1-knockout mice, the absence of ANKEF1 weakens the N-DRC's buffering capacity. This subjects the axoneme, especially during intense movement, to greater mechanical stress, leading to the deformation of B-tube and impaired sperm motility.

and destabilizing associated axonemal accessory structures (*Ghanaeian et al., 2023*; *Leung et al., 2025*; *Walton et al., 2023*).

## Discussion

During natural fertilization, males ejaculate millions of sperm; however, the majority are expelled by contractions within the female reproductive tract. Freshly ejaculated sperm display activated motility, yet only a few hundred reach the isthmus of the fallopian tube, where they bind to the mucosal epithelium to form a sperm reservoir. In this dormant state, sperm remain quiescent until ovulation occurs (*Suarez and Pacey, 2006*). Upon release from the reservoir, sperm undergo hyperactivation, enabling them to approach and penetrate the egg. Activation sperm exhibit symmetrical flagellar beats and swim along near-linear trajectories, whereas hyperactivated sperm show high-amplitude, asymmetrical flagellar movements (*Suarez and Dai, 1992*). Notably, activation is a prerequisite for hyperactivation.

The outer layer of the COC is rich in hyaluronic acid, rendering it highly viscoelastic (*Fujihara et al., 2018*). During the acrosome reaction, sperm release hyaluronidase, which degrades hyaluronic acid, softening the cumulus matrix and facilitating sperm–egg interaction. Sperm motility is crucial for penetrating the COC (*Kim et al., 2008*). Notably, vole sperm have been reported to penetrate the zona pellucida of mice and hamsters without undergoing an acrosome reaction, highlighting the role of mechanical forces in fertilization (*Wakayama et al., 1996*).

N-DRC is essential for sperm motility. However, whether additional structural components of the N-DRC remain to be identified has not yet been fully determined. As a key architectural element of the axoneme, the proper function of the N-DRC relies on intricate protein–protein interactions. ANKEF1 is an evolutionarily conserved protein characterized by a helix-turn-helix repeat structure known as the ANK domain, a motif widely implicated in mediating protein interactions (*Li et al., 2006*). Given its high expression in the testis and its identification in the TCTE1-associated LC–MS results, we hypothesized that ANKEF1 may exert a synergistic role in the functional regulation of the N-DRC.

Previous studies have demonstrated that ANKEF1 contributes to cellular adhesion and protrusion, processes critical for the morphogenesis of the *Xenopus* gastrula (*Chung et al., 2007*). Moreover, ANKEF1 expression differs between normozoospermic and asthenozoospermic males (*Zou et al., 2022*), suggesting its potential relevance to sperm function. In our study, *Ankef1*$^{-/-}$ male mice exhibited normal spermatogenesis and successfully mated and produced mating plugs, yet failed to generate offspring. Histological and morphological analyses of testes and epididymal sperm revealed no overt abnormalities. IVF results demonstrated that *Ankef1* null sperm were capable of penetrating the cumulus cell layer but failed to fertilize cumulus-intact oocytes, even following granulosa cell removal. Strikingly, fertilization was successfully achieved when ZP-free eggs were used, and embryos developed to the blastocyst stage without apparent defects. These findings indicate that ANKEF1 deficiency leads to results in male infertility due to the sperm's inability to penetrate the ZP.

Penetration of the ZP requires both a functional acrosome reaction and effective sperm motility. Treatment with calcium ionophore A23187 revealed that *Ankef1* null sperm underwent normal acrosome reactions. However, CASA showed reduced motility, suggesting that impaired ZP penetration is attributable to defective motility rather than a compromised acrosome reaction. This conclusion was further supported by sperm migration assays.

The sperm axoneme, a highly conserved '9 + 2' microtubule-based structure, is the central driver of sperm motility. Dynein arms anchored to A-tubules generate force through ATP hydrolysis, enabling sliding along adjacent B-tubules. The N-DRC is essential for converting this microtubule sliding into coordinated flagellar bending (*Satir, 1968*; *Summers and Gibbons, 1971*). Proteomic analysis identified multiple N-DRC components, including DRC5/TCTE1, DRC3/LRRC48, DRC7/CCDC135, and DRC4/GAS8. The sperm phenotype observed in ANKEF1 knockout mice closely mirrors that of TCTE1-deficient mice, suggesting that ANKEF1 may play a functionally analogous role. Supporting this, *Drc3/Lrrc48* knockout mice reach sexual maturity and exhibit normal mating behavior, but they are infertile (*Ha et al., 2016*). In Chlamydomonas, mutations in *Drc3/Lrrc48* lead to flagellar motility defects (*Awata et al., 2015*). Similarly, *Drc7/Ccdc135* knockout mice display truncated sperm tails and infertility (*Morohoshi et al., 2020*), and mutations in DRC4/GAS8 have been implicated in PCD in humans (*Jeanson et al., 2016*).

Our immunoprecipitation (IP) experiments confirmed the proteomic interactions identified by mass spectrometry, and GSEA along with ATP quantification demonstrated that ANKEF1 deletion does not impair cellular energy metabolism. These results suggest that ANKEF1 primarily contributes to the structural integrity and function of the N-DRC, independent of metabolic pathways.

Our cryo-ET and TEM results revealed that the loss of ANKEF1 causes deformation of the B-tubule in the sperm axonemal DMT structure, which may be one of the reasons for the reduced sperm motility. Recent elucidation of the 96 nm repeating unit structure of DMT in bovine sperm axonemes has significantly advanced our understanding of the spatial organization of ANKEF1 and its potential interactions with neighboring axonemal proteins (*Leung et al., 2025*). Nevertheless, the structural characterization of N-DRC remains incomplete, particularly regarding its binding interface with adjacent DMTs. High-resolution structural studies at the cellular level are still needed to define the precise architecture of the N-DRC and its connectivity to adjacent axonemal elements. Such insights would contribute to a more comprehensive understanding of the mechanistic basis of mammalian sperm motility.

Asthenospermia, defined by reduced sperm motility, represents a major cause of male infertility, yet its molecular etiology remains poorly understood. Investigating the infertility phenotype observed in *Ankef1*$^{-/-}$ mice may offer new perspectives on the underlying mechanisms of asthenozoospermia. Although current contraceptive options primarily target women, they are frequently associated with undesirable side effects such as dizziness, chest discomfort, decreased libido, and visual disturbances. Consequently, the development of safe and effective male contraceptive has become a pressing objective. Currently, no oral contraceptive is available for men, but our study offers new perspectives for male contraceptive research.

The potential of ANKEF1 as a target for male contraception arises from its essential structural role in the sperm flagellum, mediated via its ANK domain, which facilitates interaction with components of the N-DRC. Structural studies suggest that ANKEF1 features a positively charged surface, which may engage electrostatically with glutamylated tubulin in adjacent microtubules (*Leung et al., 2025*), offering a tractable interface for pharmacological targeting. Disruption of this interaction by small-molecule inhibitors could transiently impair sperm motility without affecting systemic physiology. Importantly, sperm function appears to depend more on ANKEF1 than do motile cilia in other tissues, such as those in the respiratory tract, implying that targeted inhibition of ANKEF1 would have minimal impact on ciliary function elsewhere. This type of tissue-specific phenotypic dissociation has been observed in other axonemal proteins, including DNAH17 and IQUB (*Sironen et al., 2020*; *Zhang et al., 2022*), further supporting the feasibility of ANKEF1 as a sperm-specific drug target.

# Materials and methods

## Animals

All animal experiments were approved by the Animal Care and Use Committee of the National Institute of Biological Sciences, Beijing (Approval ID: NIBS2020M0019). All procedures were performed in accordance with the committee's guidelines and relevant national regulations, and every effort was made to minimize animal suffering and the number of animals used. Age-matched adult male wild-type C57BL/6J mice (RRID:IMSR_JAX:000664) were obtained from the SPF Breeding Facility at the Animal Center of the National Institute of Biological Sciences, Beijing. The *Ankef1* knockout (*Ankef1*$^{-/-}$) and *Ankef1*-Flag knock-in mice were generated on the C57BL/6J background using CRISPR/Cas9 technology as described in the main text. Animals were housed under a controlled 12-hr light/dark cycle (lights on from 7:00 a.m. to 7:00 p.m.) with ad libitum access to standard chow and water. All experiments utilized 8- to 12-week-old male mice unless otherwise specified. This study was reported in accordance with the ARRIVE guidelines.

## Real-time quantitative PCR

Total RNA was extracted from specific tissues using TRIzol reagent (Invitrogen, 15596026) according to the manufacturer's protocol. Tissues were thoroughly homogenized using a mechanical homogenizer and incubated at room temperature for 5 min to ensure complete lysis. Chloroform was added and mixed vigorously for 15 s, followed by a 3-min incubation at room temperature. The samples were then centrifuged at 12,000×g for 15 min at 4°C (procedure repeated twice). The aqueous phase was

collected, and RNA was precipitated by adding isopropanol, gently inverting the tube several times, and incubating at room temperature for 10 min. Samples were centrifuged at $12,000 \times g$ for 10 min at 4°C, and the resulting supernatant was discarded. The RNA pellet was washed with 75% ethanol, vortexed, and centrifuged at $7500 \times g$ for 5 min at 4°C. After removing the ethanol, the pellet was air-dried for 3–5 min and resuspended in nuclease-free water. The RNA solution was incubated at 65°C for 3 min and then placed immediately on ice. Complementary DNA was synthesized using a reverse transcription kit (TaKaRa, RR047A) following the manufacturer's instructions. Quantitative PCR was performed using the SYBR Premix Ex Taq kit (TaKaRa, DRR420A), and the relative *Ankef1* expression levels were normalized to *Gapdh* as the internal control using the ΔΔCt method.

## Generation of *Ankef1*-deficient and *Ankef1*-Flag mouse

Female wild-type mouse aged 4–6 weeks was intraperitoneally injected with pregnant mare serum gonadotropin (PMSG) to stimulate follicular development, followed by injection of human chorionic gonadotropin (hCG, 10 IU/mouse) 47 hr later. After mating with male mice, fertilized zygotes were collected for subsequent microinjection. *Ankef1*$^{-/-}$ mice were generated by co-injecting Cas9 mRNA with two guide RNAs (sgRNA1 and sgRNA2) into fertilized eggs. The target sequences were as follows: sgRNA1: 5'-cctgcccactaagcggcactatc-3', sgRNA2: 5'-cctctcatgatagcgtgtgccag-3'. *Ankef1*-Flag knock-in mice were generated by co-injecting Cas9 mRNA, a guide RNA, and an *Ankef1*-Flag plasmid into fertilized embryos. Detailed gene editing strategies are provided in *Figure 1C*, *Figure 4—figure supplement 1A*.

## Histological analysis

Testes and epididymides were dissected and immediately fixed in Davidson's fixative (formaldehyde:ethanol:glacial acetic acid:distilled water = 6:3:1:10). After an initial fixation at 4°C for 2 hr, the testes were bisected with a razor blade, and tissue fragments were fully immersed in fresh fixative and incubated overnight at 4°C. The fixed tissues were then dehydrated through a graded ethanol series, embedded in paraffin, and sectioned at a thickness of 5 μm. Sections were floated on warm water at 42°C to flatten, then mounted and baked overnight at 42°C. Subsequently, the sections were deparaffinized, rehydrated, and stained with H&E following standard protocols. Histological images were acquired using an Olympus VS120 slide scanning microscope.

## IVF experiments

Female wild-type mice aged 4–6 weeks were intraperitoneally injected with PMSG to stimulate follicular development, followed 47 hr later by hCG (10 IU/mouse) to induce ovulation. Following mating with fertile males, fertilized oocytes were harvested from the females. To remove cumulus cells, oocytes were treated with hyaluronidase (Sigma-Aldrich, H3757) for 10 min. For zona pellucida removal, oocytes were treated with Tyrode's salt solution (Sigma-Aldrich, T1788) for 1 min. Sperm were introduced into TYH droplets containing either cumulus- or ZP-free oocytes, at a final concentration of $1 \times 10^6$ sperm/ml. For fertilization assays using cumulus-intact oocytes, the sperm concentration was adjusted to $1 \times 10^4$ sperm/ml. Co-incubation was performed at 37°C in an atmosphere of 5% $CO_2$, the number of embryos reaching the two-cell stage embryos was assessed after 36 hr, and blastocyst formation was evaluated after 3.5 days of culture.

## Acrosome reaction analyses

Sperm were incubated in HTF medium at 37°C in an atmosphere of 5% $CO_2$ for 30 min. A portion of the sperm was treated with the calcium ionophore A23187 (Sigma-Aldrich, 21186-5MG-F) at a final concentration of 10 μmol/l, while the control group was treated with an equivalent volume of DMSO. After 50 min of incubation, sperm were fixed with 4% PFA at room temperature for 30 min and spread onto glass slides. The slides were then stained with Coomassie Brilliant Blue solution (0.22% Coomassie Blue R-250, 50% methanol, 10% glacial acetic acid, 40% distilled water) for 10 min, followed by rinsing in distilled water to remove excess dye. Images were performed using an Olympus VS120 slide scanning microscope. For each slide, five non-overlapping regions were analyzed, with 100 sperm counted per region.

## Sperm motility analyses

Sperm were incubated in HTF medium at 37°C in 5% $CO_2$ for 1 hr. After removing cauda epididymal tissue, sperm motility was assessed using the CASA system (Version 14 CEROS, Hamilton Thorne

Research) equipped with a Slide Warmer (#720230, Hamilton Thorne Research). The acquisition settings included 30 frames captured at a frame rate of 60 Hz. Analysis parameters were as follows: minimum contrast: 30, minimum cell size: 4 pixels, and minimum static contrast = 15.

## Sperm head–tail separation

Mouse spermatozoa were subjected to repeated freeze–thaw cycles using liquid nitrogen. The samples were then centrifuged at $10,000 \times g$ for 5 min, and the resulting pellet was resuspended in 200 µl of PBS. The suspension was gently mixed with 1.8 ml of 90% Percoll solution (Sigma, P1644) and centrifuged at $15,000 \times g$ for 15 min. Following centrifugation, the sperm heads settled at the bottom of the tube, while tails remained near the top. The separated sperm heads and tails were individually diluted in PBS at a 1:5 volume ratio and centrifuged at $10,000 \times g$ for 5 min. Each fraction was then washed twice with PBS and lysed using lysis buffer (pH = 7.6, 50 mM Tris-HCl, 150 mM NaCl, 1% Triton X-100, 0.5% sodium deoxycholate, 0.1% SDS, 2 mM EDTA) supplemented with protease inhibitor cocktail (Roche, 04693116001). Subsequently, 1/5 volume of 5× loading buffer (pH = 6.8, 10% SDS, 25% glycerol, 1 M Tris-HCl, 5% β-mercaptoethanol, 0.25% bromophenol blue) was added. The mixture was boiled at 100°C for 10 min, and the supernatant was collected following centrifugation for downstream applications.

## Separation of different sperm fractions

The separation of distinct sperm fractions was performed as previously described (*Cao et al., 2006*; *Miyata et al., 2021*; *Miyata et al., 2020b*). Briefly, sperm collected from the cauda epididymis were washed twice with PBS and resuspended in 1% Triton X-100 lysis buffer (50 mM NaCl, 20 mM Tris·HCl, pH 7.5) supplemented with a protease inhibitor cocktail. The samples were incubated at 4°C for 2 hr and subsequently centrifuged at $15,000 \times g$ for 10 min. The resulting supernatant was designated as the Triton X-100-soluble fraction.

The pellet was then resuspended in 1% SDS lysis buffer (75 mM NaCl, 24 mM EDTA, pH 6.0), incubated at room temperature for 1 hr and centrifuged. The supernatant from this step was collected as the SDS-soluble fraction. The remaining pellet was boiled for 10 min in sample buffer (66 mM Tris-HCl, 2% SDS, 10% glycerol, and 0.005% bromophenol blue), followed by centrifugation. The resulting supernatant was designated as the SDS-resistant fraction. Protease inhibitor cocktail was added to both the Triton X-100 and SDS lysis buffers immediately prior to use.

## Transmission electron microscopy

TEM was conducted at the Centre for Electron Microscopy, National Institute of Biological Sciences, Beijing, according to standard protocols. Cauda epididymal tissue was excised and fixed overnight at 4°C in 2.5% glutaraldehyde (Sigma-Aldrich, G5882). Samples were then washed three times with PBS for 20 min each, followed by postfixation in 1% osmium tetroxide for 1 hr at room temperature. After another three PBS washes, the tissues were dehydrated in a graded acetone series using the progressive lowering of temperature method and subsequently embedded in epoxy resin. The embedded blocks were cured in a drying oven at 45°C for 12 hr and then at 60°C for 72 hr. Ultrathin sections were prepared and stained with 3% uranyl acetate in 70% methanol/$H_2O$, followed by Sato's lead stain for 2 min. Imaging was performed using a TECNAI spirit G2 (FEI) TEM at 120 kV.

## Western blot

Mouse tissue samples were collected and lysed in buffer (pH = 7.6; 50 mM Tris-HCl, 150 mM NaCl, 1% Triton X-100, 0.5% sodium deoxycholate, 0.1% SDS, and 2 mM EDTA) supplemented with a protease inhibitor cocktail. Prior to use, the protease inhibitors were added in appropriate proportion. The lysate was mixed with 1/5 volume of 5× loading buffer (10% SDS, 25% glycerol, 1 M Tris-HCl, 5% β-mercaptoethanol, and 0.25% bromophenol blue, pH 6.8), boiled at 100°C for 10 min, and centrifuged. The resulting supernatants were used for SDS–PAGE. Proteins were resolved by SDS–PAGE and transferred onto PVDF membranes. Membranes were blocked with 5% skim milk in TBST (20 mM Tris, 150 mM NaCl, 0.05% Tween-20, pH 7.6) for 1 hr at room temperature. Membranes were incubated with primary antibodies overnight at 4°C, followed by three washes with TBST (10 min each). Afterward, membranes were incubated with HRP-conjugated secondary antibodies at room temperature for 1 hr and washed again three times with TBST. Chemiluminescent signals were developed

using ECL reagents (Bio-Rad, 170-5060 or NCM Biotech, P10300B) and detected using XBT X-ray film (Carestream, 6535876). The antibody information used in this study is provided in *Supplementary file 2*.

## Immunofluorescence

Mouse sperm were collected in pre-warmed PBS and incubated at 37°C for 15 min. The cauda epididymis tissue was discarded, and the remaining sperm suspension was centrifuged at 1000 × *g* for 5 min. The sperm pellet was fixed in 4% PFA at room temperature for 30 min, spread on glass slides, and air-dried. Antigen retrieval was performed using antigen retrieval buffer (10 mM sodium citrate, 0.05% Tween-20, pH 6.0), followed by cooling to room temperature. Samples were then blocked with ADB blocking solution (3% BSA, 0.05% Triton X-100) for 1 hr at room temperature. Slides were incubated with the primary antibody (anti-Flag M2) overnight at 4°C and washed three times with PBST (PBS containing 0.1% Tween-20), 5 min each wash. Subsequently, slides were incubated for 1 hr at room temperature, protected from light, with secondary antibodies (Alexa Fluor 546-conjugated donkey anti-mouse IgG and Alexa Fluor 647-conjugated donkey anti-rabbit IgG) along with Hoechst 33342 (Sigma, B2261). After incubation, samples were washed three times with PBST for 5 min each. Images were acquired using a Nikon Structured Illumination Microscope (SIM) confocal system.

## Identification of interacting proteome by LC–MS

Protein bands were carefully excised from the polyacrylamide gel using a clean razor blade, ensuring minimal inclusion of excess background gel. The excised gel bands were cut into smaller fragments and transferred to 1.5 ml microcentrifuge tubes for further processing. Gel pieces were destained three times using 25 mM ammonium bicarbonate ($NH_4HCO_3$) in 50% methanol, with each step lasting 10 min. This was followed by three washes with 10% acetic acid in 50% methanol for 1 hr each. The gels were then rinsed twice with distilled water for 20 min per rinse to remove residual solvents.

After thorough rinsing, the gel pieces were transferred to 0.5 ml microcentrifuge tubes and dehydrated by adding 100% acetonitrile (ACN). Tubes were gently inverted until the gel pieces turned opaque white. The ACN was removed, and the samples were dried in a SpeedVac under mild heating for 20–30 min. For proteolytic digestion, the dried gel fragments were rehydrated with 5–20 µl of trypsin solution (10 ng/µl in 50 mM $NH_4HCO_3$, pH 8.0) and incubated at 37°C overnight.

Following digestion, peptides were extracted by incubating the gel pieces in 25–50 µl of 50% ACN with 5% formic acid (FA) for 30–60 min under gentle agitation (avoiding vortexing). The resulting supernatant was transferred to a new 0.5 ml tube. A second extraction was performed using 25–50 µl of 75% ACN with 0.1% FA under the same conditions. The supernatants from both extraction steps were combined and dried completely using the SpeedVac under gentle heat to ensure full peptide recovery for downstream LC–MS analysis.

## Verification of protein interactions in 293T cells by co-immunoprecipitation

Candidate protein-coding sequences were obtained from the NCBI database and cloned into the pcDNA3.1 expression vector. The primer sequences used for vector construction are listed in *Supplementary file 3*. Plasmids were transfected into HEK293T (ATCC, CRL-3216) cells using the jetOP-TIMUS transfection reagent (Cat. No. 101000006), and cells were incubated at 37°C in a 5% $CO_2$ atmosphere for 36 hr.

Following incubation, the culture medium was removed, and the cells were gently washed once with PBS. Cell lysis was performed using RIPA buffer (pH 7.6: 50 mM Tris-HCl, 150 mM NaCl, 1% Triton X-100 or 1% NP-40, 2 mM EDTA), supplemented with protease inhibitors as needed. A portion of the lysate was reserved as the input control, while the remaining sample was equally divided for IP using either anti-Flag or anti-Myc antibodies. The IP reactions were incubated overnight at 4°C with gentle rotation.

The immune complexes were captured using protein-conjugated beads, which were washed five times with RIPA buffer (3 min per wash). Bound proteins were eluted by adding SDS loading buffer, followed by denaturation at 95°C for 5 min. Eluted samples were subjected to SDS-PAGE and immunoblotting to assess protein–protein interactions.

## Proteomic analysis of whole sperm

Sperm samples were ground into a fine powder using liquid nitrogen and transferred into microcentrifuge tubes. Lysis buffer (8 M urea supplemented with 1% protease inhibitor cocktail) was added at a 1:4 ratio. The samples were sonicated on ice for 3 min using a Scientz high-intensity ultrasonic processor. Following sonication, the lysates were centrifuged at $12,000 \times g$ for 10 min at 4°C, and the supernatants were collected. Protein concentrations were determined using a BCA assay kit according to the manufacturer's protocol.

For protein digestion, the samples were first reduced with 5 mM DTT at 56°C for 30 min, followed by alkylation with 11 mM iodoacetamide in the dark at room temperature for 15 min. The solution was then diluted with 200 mM TEAB to reduce the urea concentration to below 2 M. Trypsin was added at a 1:50 ratio for overnight digestion, followed by a second digestion at a 1:100 ratio for 4 hr to ensure complete proteolysis. Peptides were subsequently purified using Strata X SPE columns.

For LC–MS/MS analysis, the purified peptides were dissolved in solvent A (0.1% FA and 2% ACN in water) and separated on a 25-cm in-house packed reversed-phase column (inner diameter: 100 µm). Chromatographic separation was achieved using a linear gradient of 6–80% solvent B (0.1% FA and 90% ACN) over 20 min at a flow rate of 700 nl/min using an EASY-nLC 1200 ultra-performance liquid chromatography system.

Mass spectrometry was performed on an Orbitrap Exploris 480 instrument. Full MS scans were acquired at a resolution of 60,000, and MS/MS spectra were obtained at a resolution of 15,000 using HCD with a 27% NCE. Data-independent acquisition (DIA) data were processed using DIA-NN software. Spectra were searched against the *Mus musculus* UniProt database (Mus_musculus_10090_SP_20231220.fasta), specifying Trypsin/P specified as the digestion enzyme and allowing up to one missed cleavage. Fixed modifications included N-terminal methionine excision and carbamidomethylation of cysteine residues. The false discovery rate was controlled at <1% at both the peptide and protein levels.

## Measurement of sperm ATP levels

Mouse sperm were collected in pre-warmed PBS and incubated at 37°C for 15 min. Cauda epididymal tissue was removed, and the remaining solution was centrifuged at $1000 \times g$ for 5 min. The sperm pellets were washed twice with PBS. After counting, $1 \times 10^7$ sperm cells were seeded into each well of a 96-well plate. Subsequently, 30 µl of ATP lysis (Promega, G7570) was added to each well. The plates were shaken for 20 min in the dark, and ATP levels were quantified using a MicroPlate Spectrophotometer (TECAN) according to the manufacturer's instructions.

## Mitochondrial membrane potential assessment

Sperm were collected from the cauda epididymis, and mitochondrial membrane potential was assessed using TMRM (Invitrogen, I34361) (*Dalal et al., 2016*). Briefly, mouse sperm were incubated in pre-warmed PBS at 37°C for 15 min. Following removal of the cauda tissue, the suspension was centrifuged at $300 \times g$ for 5 min, and the resulting pellets were washed twice with DPBS. The washed sperm were then incubated with TMRM for 30 min at 37°C in a humidified atmosphere containing 5% $CO_2$. After incubation, sperm were washed twice with DPBS to remove excess unbound dye. The pellet was resuspended, mixed with DAPI, and the suspension was spread onto microscope slides. Fluorescence images were acquired using a DAPI filter (excitation: 405 nm) and an RFP filter (excitation: 560 nm).

## Evaluation of intracellular ROS levels

Spermatozoa were isolated from the cauda epididymis and incubated pre-warmed TYH medium at 37°C for 50 min, with or without the ROS inducer Rosup. After incubation, samples were centrifuged gently at $300 \times g$ for 4 min and resuspended in 1 ml of DCFDA working solution. The DCFDA mix was prepared by diluting DCFH-DA (Beyotime, S0033S) at a 1:1000 ratio in TYH medium. The sperm suspension was incubated at 37°C with 5% $CO_2$ for 20 min, with gentle inversion every 5 min to ensure even dye distribution. Following incubation, the samples were centrifuged again at $300 \times g$ for 4 min, and the supernatant was discarded. The sperm pellets were then washed three times with DPBS to eliminate excess extracellular DCFH-DA. The final pellet was resuspended, mixed with DAPI, and mounted onto microscope slides. Fluorescence images were captured using a DAPI filter and an FITC filter.

## Sample preparation of mouse sperm axoneme

Freshly collected sperm were centrifuged at $400 \times g$ for 5 min at 4°C using a Thermo Scientific Legend Micro 17 R centrifuge. The resulting pellet from every 100 μl of semen was gently resuspended in 100 μl of pre-cooled PBS and subsequently diluted 5.5-fold with PBS prior to use. Cryo-EM grids (Quantifoil R3.5/1, Au 200 mesh) were glow-discharged for 60 s using a Gatan Solarus system. Sperm samples (3 μl, diluted in PBS) were applied to the grids, incubated for 3–5 s under 100% relative humidity at 4°C to allow absorption, and then the frozen samples were placed in a mixture of ethane and methane cooled to –195°C. The grids were then stored in liquid nitrogen for subsequent cryo-FIB milling. The cryo-FIB thinning strategy followed protocols described in previous work (*Tai et al., 2023*).

## Cryo-ET tilt series acquisition

The grid after cryo-FIB milling was loaded onto the Autoloader in Titan Krios G3 TEM (Thermo Fisher Scientific) operating at 300 kV. The microscope was equipped with a Gatan K2 direct electron detector and a BioQuantum energy filter. Tilt series were acquired at a magnification of ×42,000, yielding a physical pixel size of 3.4 Å in counting mode. Prior to data collection, the sample's pre-tilt was visually assessed and set to either +10° or –9°, depending on the pre-defined geometry induced during grid loading. The total electron dose per tilt was 3.5 electrons/Å², distributed across 10 frames over a 1.2-s exposure. The tilt range was –66° to +51° for a –9° pre-tilt, and –50° to +67° for a +10° pre-tilt, with 3° increments, resulting in 40 tilts per series and a cumulative dose of approximately 140 electrons/Å². The energy filter slit width was set to 20 eV, and the zero loss peak was recalibrated after the acquisition of each tilt series. The nominal defocus range was set between –1.8 and –2.5 μm. All tilt series used in this study were acquired using a beam-image shift-assisted, dose-symmetric acquisition scheme implemented with a custom SerialEM software (*Hagen et al., 2017*; *Mastronarde, 2005*; *Wu et al., 2019*).

## Mouse sperm axoneme in situ data processing

After data collection, all fractioned movies were imported into Warp for essential processing, including motion correction, 2x Fourier segmentation of super-resolution frames, CTF estimation, masking platinum islands or other high-contrast features, and tilt series generation. Subsequently, the tilt series was automatically aligned using AreTOMO (*Tegunov and Cramer, 2019*; *Zheng et al., 2022*). Aligned tilt series were visually inspected in IMOD, and any low-quality frames (such as those blocked by the sample stage or grid bars, containing obvious crystalline ice, or showing obvious jumps) were removed to create new sets of tilt series in Warp. The new tilt series underwent a second round of AreTOMO alignment. Then, low-quality frames were removed again using the same criteria as in the first round. The new tilt series was then subjected to a third round of automatic alignment, continuing the process until no frames remained to be removed. After tilt series alignment, those tilt series with fewer than 30 frames or that failed were not further processed (*Kremer et al., 1996*; *Zheng et al., 2022*). The alignment parameters of all remaining tilt series were passed back to Warp, and initial tomogram reconstruction was performed in Warp at a pixel size of 27.2 Å, resulting in a total of 160 Bin8 tomograms (*Tegunov and Cramer, 2019*).

Among the total 160 tomograms, we used the filament picking tool in Dynamo to manually pick DMT particles from 89 tomograms. By selecting the starting and ending points of each DMT fiber and separating each cutting point by 8 nm along the fiber axis, we obtained 89 sets of DMT particles (*Castaño-Díez et al., 2012*). The 3D coordinates and two of the three Euler angles (except for in-plane rotation) were automatically generated by Dynamo and then transferred back to Warp for exporting sub-tomograms (*Tegunov and Cramer, 2019*; *Castaño-Díez et al., 2012*; *Burt et al., 2021*).

In RELION 3.1, sub-tomograms were refined, using the ABTT package to transform the RELION star file and Dynamo table file and to jointly generate a mask using Dynamo and/or RELION (*Scheres, 2012*; *Zivanov et al., 2018*). First, all particles were reconstructed into a box size of $48^3$ voxels with a pixel size of 27.2 Å, and all extracted particles were directly averaged and low-pass filtered at 80 Å to generate an initial reference. Then, 3D classification with $K = 1$ was performed under the constraints of the first two Euler angles (—sigma_tilt 3 and —sigma_psi 3 in RELION), followed by 3D automatic refinement after 25 iterations. After alignment, the particles were manually cleaned in ChimeraX-1.6 (*Tai et al., 2023*; *Goddard et al., 2018*), and the aligned parameters were transferred back to Warp to export the sub-tomograms with a box size of $84^3$ voxels and a pixel size of 13.6 Å. The particles

were automatically refined in RELION. After removing any duplicate particles in Dynamo, the aligned parameters were transferred back to Warp to export the sub-tomograms with a box size of $128^3$ voxels and a pixel size of 6.8 Å. The particles were then automatically refined in RELION, resulting in a final resolution of 24 Å (*Scheres, 2012*; *Tegunov et al., 2021*; *Zivanov et al., 2018*).

## GSEA analysis

Protein mass spectrometry was performed using samples from three controls and three ANKEF1 knockout mice. Dropout data were imputed using KNN, followed by log2 normalization and sorting to obtain the final data list. Mouse gene symbols were converted to Entrez IDs using the org.Mm.eg. db annotation package, and GSEA analysis was performed with the clusterProfiler package (*Wu et al., 2021*). The term 'Glycolysis and Gluconeogenesis' was obtained from the KEGG database. Mouse gene sets were retrieved from the MSigDB (Molecular Signature Database) via the msigdbr package, focusing on category C2, which includes various biological processes and pathways. The analysis was conducted in R version 4.4.1 (2024-06-14 ucrt).

## Statistical analysis

All data are presented as the mean ± SEM ($n \geq 3$). Statistical analysis was performed using Student's *t*-test or one-way ANOVA. GraphPad Prism version 9.4.1 was used for the analysis, and results were considered significant when $p < 0.05$. Adobe Illustrator 2021 was used for image layout.

## Acknowledgements

We thank the Transgenic Animal Center at the National Institute of Biological Sciences, Beijing, for their support in generating and maintaining the transgenic mice. Additionally, we appreciate the staff at the Electron Microscopy Center and the Imaging Center of the National Institute of Biological Sciences, Beijing, for their technical support in imaging. We also extend our thanks to the State Key Laboratory for Animal Biotechnology at the College of Biological Sciences, China Agricultural University, for their help in analyzing mouse sperm motility. This work was supported by the Beijing Natural Science Foundation (JQ24056 to YZ) and the National Natural Science Foundation of China (32471244 to YZ).

## Additional information

### Funding

| Funder | Grant reference number | Author |
| --- | --- | --- |
| National Natural Science Foundation of China | 32471244 to Y.Z. | Yun Zhu |

The funders had no role in study design, data collection, and interpretation, or the decision to submit the work for publication.

### Author contributions

Shuntai Yu, Conceptualization, Data curation, Software, Formal analysis, Supervision, Validation, Investigation, Visualization, Methodology, Writing – original draft, Project administration, Writing – review and editing; Guoliang Yin, Conceptualization, Resources, Data curation, Software, Validation, Visualization, Methodology, Writing – review and editing; Peng Jin, Weilin Zhang, Xiaotong Xu, Resources; Yingchao Tian, Resources, Software; Tianyu Shao, Yushan Li, Software; Fei Sun, Supervision; Yun Zhu, Conceptualization, Resources, Software, Formal analysis, Supervision, Methodology, Project administration, Writing – review and editing; Fengchao Wang, Conceptualization, Resources, Data curation, Software, Supervision, Funding acquisition, Validation, Methodology, Project administration, Writing – review and editing

### Author ORCIDs

Shuntai Yu ![ORCID] https://orcid.org/0009-0001-7249-1502
Guoliang Yin ![ORCID] http://orcid.org/0000-0002-2156-8258

Fei Sun https://orcid.org/0000-0002-0351-5144
Yun Zhu http://orcid.org/0000-0001-9382-8592
Fengchao Wang https://orcid.org/0000-0002-3595-2859

## Ethics

All animal experiments were approved by the Animal Care and Use Committee of the National Institute of Biological Sciences, Beijing (Approval ID: NIBS2020M0019). All procedures were performed in accordance with the committee's guidelines and relevant national regulations, and every effort was made to minimize animal suffering and the number of animals used.

Reviewer #1 (Public review): https://doi.org/10.7554/eLife.105321.4.sa1
Reviewer #2 (Public review): https://doi.org/10.7554/eLife.105321.4.sa2
Author response https://doi.org/10.7554/eLife.105321.4.sa3

---

## Additional files

### Supplementary files

Supplementary file 1. Primer sequences for genotyping and RT-qPCR analysis of *Ankef1* knockout mice.

Supplementary file 2. Antibodies used in this study.

Supplementary file 3. Primer sequences for vector construction in co-immunoprecipitation (co-IP) assays.

MDAR checklist

### Data availability

Source data files have been provided for all western blots and for the mass spectrometry analyses (including the interactome data in Figure 5—source data and the quantitative proteomics data in Figure 6—source data). The cryo-electron tomography maps have been deposited in the Electron Microscopy Data Bank under accession code EMD-67587. All other data supporting the findings of this study (e.g., sperm parameters, fertility assays, histological images) are presented within the paper and its figures. The *Ankef1* knockout and *Ankef1*-Flag knock-in mouse lines and related plasmids are available from the corresponding author, Fengchao Wang (wangfengchao@nibs.ac.cn), under a materials transfer agreement.

The following dataset was generated:

| Author(s) | Year | Dataset title | Dataset URL | Database and Identifier |
|---|---|---|---|---|
| Yin G, Zhu Y, Sun F | 2025 | ANKRD5-KO Mouse Sperm Axoneme DMT | https://www.emdataresource.org/EMD-67587 | EMDataResource, EMD-67587 |

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
