## [Editor Report · eLife Assessment]

This **valuable** study reports a critical role of the axonemal protein ANKRD5 in sperm motility and male fertility. **Convincing** data were presented to support the main conclusion. This work will be of interest to biomedical researchers who study ciliogenesis, sperm biology, and male fertility.

---

## [Referee Report · Reviewer #1 (Public review)]

Summary:

Asthenospermia, characterized by reduced sperm motility, is one of the major causes of male infertility. The "9 + 2" arranged MTs and over 200 associated proteins constitute the axoneme, the molecular machine for flagellar and ciliary motility. Understanding the physiological functions of axonemal proteins, particularly their links to male infertility, could help uncover the genetic causes of asthenospermia and improve its clinical diagnosis and management. In this study, the authors generated Ankrd5 null mice and found that ANKRD5-/- males exhibited reduced sperm motility and infertility. Using FLAG-tagged ANKRD5 mice, mass spectrometry, and immunoprecipitation (IP) analyses, they confirmed that ANKRD5 is localized within the N-DRC, a critical protein complex for normal flagellar motility. However, transmission electron microscopy (TEM) and cryo-electron tomography (cryo-ET) of sperm from Ankrd5 null mice did not reveal significant structural abnormalities.

Strengths:

The phenotypes observed in ANKRD5-/- mice, including reduced sperm motility and male infertility, are conversing. The authors demonstrated that ANKRD5 is an N-DRC protein that interacts with TCTE1 and DRC4. Most of the experiments are well-designed and executed.

Comments on revised version:

My concerns have been addressed.

---

## [Referee Report · Reviewer #2 (Public review)]

Summary:

The manuscript investigates the role of ANKRD5 (ANKEF1) as a component of the N-DRC complex in sperm motility and male fertility. Using Ankrd5 knockout mice, the study demonstrates that ANKRD5 is essential for sperm motility and identifies its interaction with N-DRC components through IP-mass spectrometry and cryo-ET. The results provide insights into ANKRD5's function, highlighting its potential involvement in axoneme stability and sperm energy metabolism.

Strengths:

The authors employ a wide range of techniques, including gene knockout models, proteomics, cryo-ET, and immunoprecipitation, to explore ANKRD5's role in sperm biology.

Comments on revised version:

The authors have already addressed the issues I am concerned about.

---

## [Author Response]

The following is the authors’ response to the previous reviews

**Reviewer #1 (Public review):**
Summary:Asthenospermia, characterized by reduced sperm motility, is one of the major causes of male infertility. The "9 + 2" arranged MTs and over 200 associated proteins constitute the axoneme, the molecular machine for flagellar and ciliary motility. Understanding the physiological functions of axonemal proteins, particularly their links to male infertility, could help uncover the genetic causes of asthenospermia and improve its clinical diagnosis and management. In this study, the authors generated Ankrd5 null mice and found that ANKRD5-/- males exhibited reduced sperm motility and infertility. Using FLAG-tagged ANKRD5 mice, mass spectrometry, and immunoprecipitation (IP) analyses, they confirmed that ANKRD5 is localized within the N-DRC, a critical protein complex for normal flagellar motility. However, transmission electron microscopy (TEM) and cryo-electron tomography (cryo-ET) of sperm from Ankrd5 null mice did not reveal significant structural abnormalities.Strengths:The phenotypes observed in ANKRD5-/- mice, including reduced sperm motility and male infertility, are conversing. The authors demonstrated that ANKRD5 is an N-DRC protein that interacts with TCTE1 and DRC4. Most of the experiments are well designed and executed.Weaknesses:The last section of cryo-ET analysis is not convincing. "ANKRD5 depletion may impair buffering effect between adjacent DMTs in the axoneme"."In WT sperm, DMTs typically appeared circular, whereas ANKRD5-KO DMTs seemed to be extruded as polygonal. (Fig. S9B,D). ANKRD5-KO DMTs seemed partially open at the junction between the A- and B-tubes (Fig. S9B,D)." In the TEM images of 4E, ANKRD5-KO DMTs look the same as WT. The distortion could result from suboptimal sample preparation, imaging or data processing. Thus, the subsequent analyses and conclusions are not reliable.

Thank you for your valuable advice. To validate the results of cryo-ET, we carefully analyzed the TEM results (previously we only focused on the global "9+2" structure of the axial filament) and found that deletion of ANKRD5 resulted in both normal and deformed DMT morphologies, which was consistent with the results observed by cryo-ET. At the same time, we have added the corresponding text and picture descriptions in the article:

The text description we added is: “Upon re-examining the TEM data in light of the Cryo-ET findings, similar abnormalities were observed in the TEM images (Fig.4E, Fig. S10B). Notably, both intact and deformed DMT structures were consistently observed in both TEM and STA analyses, with the deformation of the B-tube being more obvious (Fig.4E, Fig. S10). ”

This paper still requires significant improvements in writing and language refinement. Here is an example: "While N-DRC is critical for sperm motility, but the existence of additional regulators that coordinate its function remains unclear" - ill-formed sentences.

We appreciate the reviewer’s valuable comment regarding the clarity of our writing. The sentence cited (“While N-DRC is critical for sperm motility, but the existence of additional regulators that coordinate its function remains unclear”) was indeed ill-formed. We have revised it to improve readability and precision. The corrected version now reads:“Although the N-DRC is critical for sperm motility, whether additional regulatory components coordinate its function remains unclear.” We have carefully re-examined the manuscript and refined the language throughout to ensure clarity and conciseness.

**Reviewer #2 (Public review):**
Summary:The manuscript investigates the role of ANKRD5 (ANKEF1) as a component of the N-DRC complex in sperm motility and male fertility. Using Ankrd5 knockout mice, the study demonstrates that ANKRD5 is essential for sperm motility and identifies its interaction with N-DRC components through IP-mass spectrometry and cryo-ET. The results provide insights into ANKRD5's function, highlighting its potential involvement in axoneme stability and sperm energy metabolism.Strengths:The authors employ a wide range of techniques, including gene knockout models, proteomics, cryo-ET, and immunoprecipitation, to explore ANKRD5's role in sperm biology.Weaknesses:“Limited Citations in Introduction: Key references on the role of N-DRC components (e.g.,DRC2, DRC4) in male infertility are missing, which weakens the contextual background.”

We appreciate the reviewer’s valuable suggestion. To address this concern, we have added the following sentence in the Introduction:

“Recent mammalian knockout studies further confirmed that loss of DRC2 or DRC4 results in severe sperm flagellar assembly defects, multiple morphological abnormalities of the sperm flagella (MMAF), and complete male infertility, highlighting their indispensable roles in spermatogenesis and reproduction [31].”

This addition introduces up-to-date evidence on DRC2 and DRC4 functions in male infertility and strengthens the contextual background as recommended.

**Reviewer #1 (Recommendations for the authors):**
"Male infertility impacts 8%-12% of the global male population, with sperm motility defects contributing to 40%-50% of these cases [2，3]. " Is reference 3 proper? I don't see "sperm motility defects contributing to 40%-50%" of male infertility.

Thank you for identifying this issue. You are correct—reference 3 does not support the statement about sperm motility defects comprising 40–50% of male infertility cases; it actually states:

“Male factor infertility is when an issue with the man’s biology makes him unable to impregnate a woman. It accounts for between 40 to 50 percent of infertility cases and affects around 7 percent of men.”

This was a misunderstanding on my part, and I apologize for the oversight.

To correct this, we have replaced the statement with more accurate references:

PMID: 33968937 confirms:

“Asthenozoospermia accounts for over 80% of primary male infertility cases.”

PMID: 33191078 defines asthenozoospermia (AZS) as reduced or absent sperm motility and notes it as a major cause of male infertility.

We have updated the manuscript accordingly:

In the Significance Statement: “Male infertility affects approximately 8%-12% of men globally, with defects in sperm motility accounting for over 80% of these cases.”

In the Introduction: “Male infertility affects approximately 8% to 12% of the global male population, with defects in sperm motility accounting for over 80% of these cases[2,3].”

Thank you again for your careful review and for giving us the opportunity to improve the accuracy of our manuscript.

"Rather than bypassing the issue with ICSI, infertility from poor sperm motility could potentially be treated or even cured through stimulation of specific signaling pathways or gene therapy." Need references.

We appreciate the reviewer’s insightful comment. In response, we have added three supporting references to the relevant sentence.

The first reference (PMID: 39932044) demonstrates that cBiMPs and the PDE-10A inhibitor TAK-063 significantly and sustainably improve motility in human sperm with low activity, including cryopreserved samples, without inducing premature acrosome reaction or DNA damage. The second reference (PMID: 29581387) shows that activation of the PKA/PI3K/Ca²⁺ signaling pathways can reverse reduced sperm motility. The third reference (PMID: 33533741) reports that CRISPR-Cas9-mediated correction of a point mutation in Tex11^PM/Y^ spermatogonial stem cells (SSCs) restores spermatogenesis in mice and results in the production of fertile offspring.

These references provide mechanistic support and demonstrate the feasibility of treating poor sperm motility through targeted pathway modulation or gene therapy, thus reinforcing the validity of our statement.

"Our findings indicate that ANKRD5 (Ankyrin repeat domain 5; also known as ANK5 or ANKEF1) interacts with N-DRC structure". The full name should be provided the first time ANKRD5 appears. Is ANKRD5 a component of N-DRC or does it interact with N-DRC?

We thank the reviewer for the valuable suggestion. In response, we have moved the full name “Ankyrin repeat domain 5; also known as ANK5 or ANKEF1” to the abstract where ANKRD5 first appears, and have removed the redundant mention from the main text.

Based on our experimental data, we consider ANKRD5 to be a novel component of the N-DRC (nexin-dynein regulatory complex), rather than merely an interacting partner. Therefore, we have revised the sentence in the main text to read:

“Here, we demonstrate that ANKRD5 is a novel N-DRC component essential for maintaining sperm motility.”

Fig 5E, numbers of TEM images should be added.

We thank the reviewer for the suggestion. We would like to clarify that Fig. 5E does not contain TEM images, and it is likely that the reviewer was referring to Fig. 4E instead.

In Fig. 4E, we conducted three independent experiments. In each experiment, 60 TEM cross-sectional images of sperm tails were analyzed for both Ankrd5 knockout and control mice.

The findings were consistent across all replicates.

We have updated the figure legend accordingly, which now reads:

“Transmission electron microscopy (TEM) of sperm tails from control and Ankrd5 KO mice. Cross-sections of the midpiece, principal piece, and end piece were examined. Red dashed boxes highlight regions of interest, and the magnified views of these boxed areas are shown in the upper right corner of each image. In three independent experiments, 20 sperm cross-sections per mouse were analyzed for each group, with consistent results observed.”

There are random "222" in the references. Please check and correct.

I sincerely apologize for the errors caused by the reference management software, which resulted in the insertion of random "222" and similar numbering issues in the reference list. I have carefully reviewed and corrected the following problems:

References 9, 11, 13, 26, 34, 63, and 64 had the number "222" mistakenly placed before the title; these have now been removed. References 15 and 18 had "111" incorrectly inserted before the title; this has also been corrected. Reference 36 had an erroneous "2" before the title and was found to be a duplicate of Reference 32; these have now been merged into a single citation. Additionally, References 22 and 26 were identified as duplicates of the same article and have been consolidated accordingly.

All these issues have been resolved to ensure the reference list is accurate and properly formatted.

**Reviewer #2 (Recommendations for the authors):**
The authors have already addressed most of the issues I am concerned about.

In addition, we have also corrected some errors in the revised manuscript:

(1) In Figure 3G, the y-axis label was previously marked as “Sperm count in the oviduct (10⁶)”, which has now been corrected to “Sperm count in the oviduct”.

(2) All p-values have been reformatted to italic lowercase letters to comply with the journal style guidelines.

Figure 6 Legend: A typographical error in the figure legend has been corrected. The text previously read “(A) The differentially expressed proteins of Ankrd5^+/–^ and Ankrd5^+/-^ were identified...”. This has now been amended to “(A) The differentially expressed proteins of Ankrd5^+/–^ and Ankrd5^+/–^ were identified...” to correctly represent the comparison between heterozygous and homozygous knockout groups.

In the original Figure 4E, we added a zoom-in panel to the image to show the deformed DMT.